

# Modelling wet snow avalanche runout to assess road safety at a high-altitude mine in the central Andes

Cesar VERA VALERO[1], Nander WEVER[2], Yves BÜHLER[1], Lukas STOFFEL[1], Stefan MARGRETH[1], and Perry BARTELT[1]

[1]WSL Institute for Snow and Avalanche Research SLF, Flüelastrasse 11, 7260 Davos Dorf, Switzerland
[2]École Polytechnique Fédérale de Lausanne (EPFL), School of Architecture, Civil and Environmental Engineering, Lausanne, Switzerland

*Correspondence to:* C. Vera (cesar.vera@slf.ch)

**Abstract.** Mining activities in cold regions are vulnerable to snow avalanches. Unlike operational facilities, which can be constructed in secure locations outside the reach of avalanches, access roads are often susceptible to being cut, leading to mine closures and significant financial losses. In this paper we discuss the application of avalanche runout modelling to predict the operational risk to mining roads, a longstanding problem for mines in high-altitude, snowy regions. We study the 35 km long road located in the "Cajon del Rio Blanco" valley in the central Andes which is operated by the Codelco Andina copper mine. In winter and early spring this road is threatened by over 100 avalanche paths. If the release and snowcover conditions can be accurately specified, we find that avalanche dynamics modelling is able to represent runout and safe traffic zones can be identified. We apply a detailed, physics based snowcover model to predict snow temperature, density and moisture content in three-dimensional terrain. This information is used to determine the initial and boundary conditions of the avalanche dynamics model. Of particular importance is the assesment of the current snow conditions along the avalanche tracks which define the mass and thermal energy entrainment rates and therefore the possibility of avalanche growth and long runout distances.

## 1 Introduction

High-altitude mining activities are frequently disrutped by snow avalanches. Historically, three of the most severe avalanche disasters ever recorded have destroyed mining settlements. On December 5th, 1935, a large avalanche released from Mount Iukspor and destroyed wooden buildings constructed to house workers of a Soviet apatite mine in the Khibiny mountains. The avalanche killed 89 people (Bruno, 2013). On the 8th of August, 1944, the *Teniente* copper mine in the central Andes



was struck by a catastrophic avalanche, killing more than 100 workers in the Sewell mining camp (Vergara and Baros, 2002; León, 2003). The worst mining avalanche disaster occurred on February 9th, 1945, when an avalanche buried the living quarters of the coal mine *October* on Sakhalin island, killing 131 people (Podolskiy et al., 2014). The avalanche disasters in the Khibinys and Sakhalin

are of great historical importance since they motivated avalanche studies in the former Soviet Union (Bruno, 2013).

Nowdays the majority of mine workers do not live in mining camps close to the operation areas. Mines are operated in shifts where a large number of workers are transported in and out of the primary excavation areas. The main risk from avalanches occurs during shift changes when miners

are exposed to avalanche danger on access roads. The miners are transported in long bus convoys containing many vehicles and therefore are at great risk. During high avalanche risk periods the access roads must be closed; causing significant financial losses because mine operations and shift changes are disrupted.

For this reason large mines have well-trained avalanche winter operation crews who are respon-

sible for road management. The winter operation crews must make closure decisions often well in advance of avalanche activity in order to plan the next operational shift. Safety experts therefore require methods to assess avalanche danger. They use automatic weather stations and have some data on the current snowcover conditions, including snowpit measurements. However, unlike avalanche forecasters in ski regions, the primary question avalanche experts in mines must answer is directly

related to road traffic; that is, can avalanches reach the road? A secondary question then arises: if the road is buried by an avalanche, how quickly can it be cleared and reopened? Safety crews can position clearing equipment in different locations according to where they expect the largest avalanche deposits in order to open the roads as quickly as possible, minimizing the operational disruption. These questions involve both the problem of snowcover stability and the problem of

expected avalanche runout.

In this paper we discuss the use of avalanche dynamics models that use initial input data defined by current snowcover conditions. At this stage of the investigation, the goal is to determine the quality of the dynamic modeling to accurately and consistently predict avalanche runout, and not yet, if ever, to define real time hazard maps. Our goal is to identify how accurate initial conditions must

be defined (snow release height, temperature and moisture content) in order to make reliable runout predictions. Model comparison to observations is a first step to integrating avalanche dynamics calculations in an operational environment. The problem is of great interest, because it requires the simulation of small, frequent avalanches, a task which is increasingly arising in engineering offices, but one that represents a large change in the application of traditional avalanche dynamics models.

Recent advances in snow avalanche dynamics research make this work possible. For one, the mean avalanche temperature has been introduced as an independent state variable in avalanche calculations (Vera et al., 2015). Avalanche temperature is controlled by the temperature of the snow at





release as well as by the temperature of the snow entrained along the path. Moreover, not only is mass entrained, but also its thermal energy. Although it is well-known that avalanche flow regime

is a function of snow temperature, (see e.g. Bozhinskiy and Losev (1998); Gauer et al. (2008); Issler and Gauer (2008); Steinkogler et al. (2014)), it is only recently that a statistical correlation between temperature and avalanche runout has been established (Naaim et al., 2013). Modelling how the temperature affects avalanche runout requires postulating temperature dependent functions for avalanche friction. The long runout distances of wet avalanches suggest a decrease in Coulomb fric-

tion induced by lubricated gliding at the basal boundary which controls the reach of the avalanche. This fact was recognized early by Voellmy, who postulated that Coulomb friction decreased to zero either by fluidization *or* by meltwater lubrication (Voellmy, 1955). Experimental field measurements indicate wet snow flows exhibit slower, plug-like velocity profiles where shearing is concentrated at the avalanche base (Dent et al., 1998; Kern et al., 2009). Isothermal, moist snow is typically associ-

ated with dense flows in the frictional flow regime indicating that velocity fluctuations are strongly damped with increasing snow temperature (Buser and Bartelt, 2015). This serves to concentrate the dissipation within a thin shear layer located at the base of the avalanche, concentrating the frictional heating (and therefore the meltwater production) at the running surface (Miller et al., 2003). Another effect is the increase of snow cohesion with increasing temperature (Voytokskiy, 1977), further pre-

venting the fluidization of the avalanche core and the transition to fluidized flow regimes (Bozhinskiy and Losev, 1998; Bartelt et al., 2015).

   To demonstrate how initial and boundary conditions control avalanche flow, we simulate several avalanches documented during three winter field campaigns at the "Cajón del rio Blanco" Valley of the Codelco Andina mine situated 100 kilometers North East from Santiago in the Chilean An-

des. This region is well-known for wet snow avalanche activity (León, 2003; McClung, 2013). The terrain is represented using a 2m high resolution DEM (digital elevation model). To model the observed avalanches, we employ an avalanche dynamics model that tracks the mean avalanche temperature, including the production of meltwater from frictional dissipation. We postulate a meltwater dependent lubrication function and investigate the sensitivity of the simulations to temperature and

meltwater. To estimate the snowcover conditions at the avalanche release and erosion areas, numerical snow cover simulations using the detailed, physics based SNOWPACK model were used (Bartelt et al., 2002; Lehning et al., 2002), driven by meteorological data from automatic weather stations over a period of five winter seansons. The SNOWPACK model results were validated with field measurements (snow pits) performed by the winter operation crew. An additional problem is the

danger arising from small point releases, often containing only 100 m$^3$ of mass. Avalanche growth by entrainment is therefore critical to model runout and the final deposition volume.

   The results indicate that avalanche runout forecasting applications might be possible in the near future if accurate snow cover information, coupled with high resolution terrain models, can be used to drive avalanche dynamics calculations. Such tools could significantly support the existing exper-



tise and know-how of mine road safety crews.

## 2  Model

### 2.1  Model Equations

Avalanche activity in the central Andes is dominated by the Pacific maritime climate of Chile (León,
2003). Extreme precipitation events are often followed by intense warming leading to the formation
of wet snow avalanches, especially in the late winter months of August and September. Another
feature of the Chilean Andes is the relatively high elevation of the release zones (between 2500 m
and 4500 m). This leads to avalanches that often start in sub-zero temperatures and run into moist,
isothermal snowcovers. Sub-zero release areas can lead to formation of dry mixed flowing/powder
type avalanches that transition at lower elevations to moist, wet flows. Another distinctive feature
of the high altitude slopes is the absence of vegetation. The sliding surfaces are mostly bedrock and
rock scree, see Fig. 1.

Simulation of avalanches in this environment requires a general avalanche dynamics model that
accounts for both collisional, (powder) and frictional (wet) flow regimes. The model assumes that
the flowing avalanche core consists of mass in the form of snow particles and clods that are created
when the snowcover is set in motion (Fig. 2). In the following we will model only the core $\Phi$ of the
avalanche, the air blast from the motion of the dust cloud will not be considered.

The particles within the core consist of snow and can contain some water (Fig. 2). The three
sources of water are (1) meltwater in the release zone, (2) moist snow entrained by the avalanche or
(3) meltwater produced by frictional heating during the motion of the avalanche. The density of the
individual snow particles is large ,we take a mean granule density to be $\rho_g = 450$ kg m$^{-3}$, (Jomelli
and Bertran, 2001; Bartelt and McArdell, 2009), but the particles can disperse leading to smaller
bulk avalanche flow densities.

Letting $M_\Phi^s$ denote the snow mass per unit area of the running surface and $M_\Phi^w$ designating the
total water mass per unit area, the total mass of the avalanche is $M_\Phi$,

$$M_\Phi = M_\Phi^s + M_\Phi^w = \rho_s h_\Phi^s + \rho_w h_\Phi^w = \rho_i h_\Phi^i + \rho_a h_\Phi^a + \rho_w h_\Phi^w \qquad (1)$$

where $\rho_s$ is the density of the flowing snow; $\rho_i$, $\rho_w$, $\rho_a$ are the densities of ice, water and air,
respectively. The height of the flowing snow is denoted $h_\Phi^s$ and the total height (volume per unit
area) of meltwater is denoted $h_\Phi^w$. When $M_\Phi^w = 0$, the avalanche is termed 'dry'; 'wet' flows occur
when $M_\Phi^w > 0$. The volumetric contents of ice, air and water in the flowing avalanche core are,

$$\theta_\Phi^i = h_\Phi^i/h_\Phi \qquad \theta_\Phi^i = h_\Phi^i/h_\Phi \qquad \theta_\Phi^i = h_\Phi^i/h_\Phi. \qquad (2)$$

We will assume that the mass of water is transported within the travelling snow which is moving
in the slope parallel direction with velocity $\mathbf{u}_\Phi = (u_\Phi, v_\Phi)^T$, see Fig. 2. The meltwater is therefore





bonded to the snow mass, filling the pore space of the particles. Typical liquid water contents (LWC) values range between $0 \leq$ and $\leq 15\%$ in individual layers within the snowcover and rarely exceed 5% in average (Heilig et al., 2015). The water content of the granules defines the nature of the frictional interaction between the particles and the sliding surface. No water mass can be lost to the sliding surface, as we always remain in the capillary regime and therefore have large capillary pressures holding the water in the particles.

In the following we make the additional assumption that a column of mass in the avalanche can expand vertically (Fig. 3), changing the flow density of the avalanche core (Buser and Bartelt, 2015). Shearing in the avalanche core $\mathbf{S}_\Phi$ induces particle trajectories that are no longer in line with the mean downslope velocities $\mathbf{u}_\Phi$ (Gubler, 1987; Bartelt et al., 2006). The kinetic energy associated with the velocity fluctuations is denoted $R_\Phi^K$. The basal boundary plays a prominent role because particle motions in the slope-perpendicular direction are inhibited by the boundary and reflected back into the flow. The basal boundary converts the production of random kinetic energy $R_\Phi^K$ into an energy flux that changes the $z$-location of particles and therefore the potential energy and particle configuration within the avalanche core. The potential energy of the configuration of the particle ensemble is denoted $R_\Phi^V$. The center-of-mass of the granular ensemble moves with the slope perpendicular velocity $w_\Phi$. When $w_\Phi > 0$, the granular ensemble is expanding; conversely when $w_\Phi < 0$, the volume is contracting (Fig. 3).

The densest packing of granules defines the co-volume height $^0h_\Phi^s$ and density $^0\rho_\Phi^s$. The co-volume has the property that $h_\Phi^s \geq ^0 h_\Phi^s$ and $\rho_\Phi^s \leq ^0 \rho_\Phi^s$. An excess pressure is created at the basal boundary when the volume expands. This pressure is termed the dispersive pressure and denoted $N_K$. It is the reaction at the base to the upward acceleration of the granular ensemble,

$$N_K = M_\Phi \dot{w}_\Phi. \tag{3}$$

The acceleration along the slope's perpendicular direction is denoted $g'$ and is composed of the slope perpendicular component of gravity $g_z$, dispersive acceleration $\dot{w}_\Phi$ and centripetal accelerations $f_z$, (Fischer et al., 2012). The total normal force at the base of the avalanche is given by $N$,

$$N = M_\Phi g' = M_\Phi g_z + N_K + M_\Phi f_z. \tag{4}$$

The sum of the random kinetic $R_\Phi^K$ and configurational energies $R_\Phi^V$, that is the potential energy resulting from a volume increase, is called the free mechanical free energy of the avalanche $R_\Phi$,

$$R_\Phi = R_\Phi^K + R_\Phi^V. \tag{5}$$

The production of free mechanical energy $\dot{P}_\Phi$, (the notation $(\dot{\ })$ means time derivative) is given by an equation containing two model parameters: the production parameter $\alpha$ and the decay parameter $\beta$, see (Buser and Bartelt, 2009)

$$\dot{P}_\Phi = \alpha [\mathbf{S}_\Phi \cdot \mathbf{u}_\Phi] - \beta R_\Phi^K h_\Phi. \tag{6}$$





The production parameter $\alpha$ defines the generation of the total free mechanical energy from the shear work rate $[\mathbf{S}_\Phi \cdot \mathbf{u}_\Phi]$; the parameter $\beta$ defines the decrease of the kinetic part $R_\Phi^K$ by inelastic particle interactions. Snow temperature and liquid water content have a strong influence on the mechanical properties of snow and therefore the amount of free mechanical energy in the avalanche. The primary difference between wet and dry flows is the production and dissipation of free mechanical energy, which controls the fluidization of the avalanche core. When the avalanche snow contains some free water, the hardness of the granules decreases (Voytokskiy, 1977), and they can be plastically deformed and sculptured into well-rounded forms (Bozhinskiy and Losev, 1998).

The energy flux associated with the configurational changes is denoted $\dot{P}_\Phi^V$ and given by

$$\dot{P}_\Phi^V = \gamma \dot{P}_\Phi. \tag{7}$$

The parameter $\gamma$ therefore determines the magnitude of the dilatation of the flow volume under a shearing action. When $\gamma = 0$ there is no volume expansion by shearing. Therefore, the model formulation we apply allows the simulation of both disperse and dense avalanche flow types. In this paper we are primarily concerned with dense, plug-like wet snow avalanche movements with no volume increase ($R_\Phi^V \approx 0$); however, as we shall see in the case studies, even wet flows fluidize in steep, rough terain ($R_\Phi^V > 0$, $\gamma > 0$). We model this material property by using production coefficients $\alpha \geq 0.05$ and large free mechanical energy decay coefficients $\beta > 1.0$ for wet snow (Buser and Bartelt, 2009). This ensures that only in very rough and steep terrain fluidization of the wet avalanche core is possible. Runout on flat slopes is therefore governed by lubrication.

Frictional heating is concentrated on the particle surfaces where the shearing and collisional and rubbing interactions occur (Voellmy, 1955; Miller et al., 2003), see Fig. 2. Although we expect temperature differences between the interior and exterior layers of the granules, as well as temperature distributions in the flow depth, we model temperature dependent effects by tracking the depth-averaged avalanche temperature $T_\Phi$ within the flow (Vera et al., 2015). The temperature $T_\Phi$ is related to the internal heat energy $E_\Phi$ by the specific heat capacity of snow $c_\Phi$

$$E_\Phi = \rho_\Phi c_\Phi T_\Phi. \tag{8}$$

The avalanche temperature is governed by (1) the initial temperature of the snow $T_0$, (2) dissipation of kinetic energy by shearing $\dot{Q}_\Phi$, as well as (3) thermal energy input from entrained snow $\dot{Q}_{\Sigma \to \Phi}$ and (4) latent heat effects from phase changes $\dot{Q}_w$ (meltwater production), see (Vera et al., 2015). Dissipation is the part of the shear work not being converted into free mechanical energy in addition to the inelastic interactions between particles that is the decay of random kinetic energy, $R_\Phi^K$

$$\dot{Q}_\Phi = (1 - \alpha)[\mathbf{S}_\Phi \cdot \mathbf{u}_\Phi] + \beta R_\Phi^K h_\Phi. \tag{9}$$

In summary, the flow of the avalanche core is described by nine state variables:

$$\mathbf{U}_\Phi = (M_\Phi, M_\Phi u_\Phi, M_\Phi v_\Phi, R_\Phi h_\Phi, E_\Phi h_\Phi, h_\Phi, M_\Phi w_\Phi, N_K, M_w)^T. \tag{10}$$



The model equations can be conveniently written as a single vector equation:

$$\frac{\partial \mathbf{U}_\Phi}{\partial t} + \frac{\partial \mathbf{\Phi}_x}{\partial x} + \frac{\partial \mathbf{\Phi}_y}{\partial y} = \mathbf{G}_\Phi \tag{11}$$

where the components $(\mathbf{\Phi}_x, \mathbf{\Phi}_y, \mathbf{G}_\Phi)$ are:

$$\mathbf{\Phi}_x = \begin{pmatrix} M_\Phi u_\Phi \\ M_\Phi u_\Phi^2 + \frac{1}{2} M_\Phi g' h_\Phi \\ M_\Phi u_\Phi v_\Phi \\ R_\Phi h_\Phi u_\Phi \\ E_\Phi h_\Phi u_\Phi \\ h_\Phi u_\Phi \\ M_\Phi w_\Phi u_\Phi \\ N_K u_\Phi \\ M_w u_\Phi \end{pmatrix}, \ \mathbf{\Phi}_y = \begin{pmatrix} M_\Phi v_\Phi \\ M_\Phi u_\Phi v_\Phi \\ M_\Phi v_\Phi^2 + \frac{1}{2} M_\Phi g' h_\Phi \\ R_\Phi h_\Phi v_\Phi \\ E_\Phi h_\Phi v_\Phi \\ h_\Phi v_\Phi \\ M_\Phi w_\Phi v_\Phi \\ N_K v_\Phi \\ M_w v_\Phi \end{pmatrix}, \ \mathbf{G}_\Phi = \begin{pmatrix} \dot{M}_{\Sigma \to \Phi} \\ G_x - S_{\Phi x} \\ G_y - S_{\Phi y} \\ \dot{P}_\Phi \\ \dot{Q}_\Phi + \dot{Q}_{\Sigma \to \Phi} + \dot{Q}_w \\ w_\Phi \\ N_K \\ 2\gamma \dot{P}_\Phi - 2N w_\Phi / h_\Phi \\ \dot{M}_{\Sigma \to w} + \dot{M}_w \end{pmatrix}. \tag{12}$$

The mathematical description of mountain terrain is defined using a horizontal $X$-$Y$ coordinate system. The elevation $Z(X,Y)$ is specified for each $(X,Y)$ coordinate pair. This information is used to define the local surface $(x,y,z)$ coordinate system with the directions $x$ and $y$ parallel to the geographic coordinates $X$ and $Y$. The grid of geographic coordinates defines inclined planes with known orientation; the $z$-direction is defined perpendicular to the local $x$-$y$ plane. The flowing avalanche is driven by the gravitational acceleration in the tangential directions $\mathbf{G} = (G_x, G_y) = (M_\Phi g_x, M_\Phi g_y)$. The model equations are solved using the same numerical schemes outlined in Christen et al. (2010). The derivation from the thermal energy and vertical motion equations are presented at Vera et al. (2015); Buser and Bartelt (2015).

## 2.2 Entrainment of warm, moist snow

We treat the entrainment of warm, moist snow as a fully plastic collision between the avalanche core $\Phi$ and snow cover $\Sigma$. By definition of a plastic collision, entrained snow is initially at rest, but after the collision with the avalanche all the entrained mass is moving with the avalanche velocity $\mathbf{u}_\Phi$. A layer of snow with height $l_\Sigma$, density $\rho_\Sigma$ and temperature $T_\Sigma$ is entrained at the rate $\dot{M}_{\Sigma \to \Phi}$ (Fig. 2). If the entrained snow is moist, in addition to the snow mass, water mass is entrained at the rate $\dot{M}_{\Sigma \to w}$. The entrained mass is composed of ice (superscript $i$), water (superscript $w$) and air (superscript $a$),

$$M_{\Sigma \to \Phi} = \rho_\Sigma l_\Sigma = \rho_i l_\Sigma^i + \rho_a l_\Sigma^a + \rho_w l_\Sigma^w. \tag{13}$$

The rate the snowcover is being eroded $\dot{l}_\Sigma$ is defined by the dimensionless erodibility coefficient $\kappa$ (Christen et al., 2010),

$$\dot{l}_\Sigma = \kappa \| \mathbf{u}_\Phi \|. \tag{14}$$



The wet and dry components of the snowcover are entrained at the same rate, proportional to the volumetric components of the snow layer

$$\dot{l}_\Sigma^i = \theta_\Sigma^i \dot{l}_\Sigma \quad \dot{l}_\Sigma^w = \theta_\Sigma^w \dot{l}_\Sigma \quad \dot{l}_\Sigma^a = \theta_\Sigma^a \dot{l}_\Sigma \tag{15}$$

where $\theta$ is the volumetric component of ice, water and air, $\theta_\Sigma^i = l_\Sigma^i / l_\Sigma$, etc. The total snow mass that is entrained is

$$\dot{M}_{\Sigma \to \Phi} = \rho_\Sigma \kappa \|\mathbf{u_\Phi}\|. \tag{16}$$

The entrained water mass is therefore,

$$\dot{M}_{\Sigma \to w} = \theta_\Sigma^w \dot{M}_{\Sigma \to \Phi}. \tag{17}$$

The thermal energy entrained during the mass intake is

$$\dot{Q}_{\Sigma \to \Phi} = \left[ \theta_\Sigma^i c_i + \theta_\Sigma^w c_w + \theta_\Sigma^a c_a + \frac{1}{2} \frac{\|\mathbf{u_\Phi}\|^2}{T_\Sigma} \right] \dot{M}_{\Sigma \to \Phi} T_\Sigma \tag{18}$$

where $c_i$, $c_w$ and $c_a$ are the specific heat capacity of ice, water and air, respectively. When the snow layer contains water $\theta_\Sigma^w > 0$, then the temperature of the entire layer is set to $T_\Sigma = 0°$ C. Equation 18 takes into account the production of heat energy during the plastic collision. In this entrain-
ment model no random kinetic energy is generated because the entrainment process is considered a perfectly plastic collision.

### 2.3   Wet snow avalanche flow rheology

Wet snow avalanches are regarded as dense granular flows in the frictional flow regime (Voellmy, 1955; Bozhinskiy and Losev, 1998). Measured velocity profiles exhibit pronounced visco-plastic
like character and are often modelled with a Bingham-type flow rheology (Dent and Lang, 1983; Norem et al., 1987; Salm, 1993; Dent et al., 1998; Bartelt et al., 2005; Kern et al., 2009). Granules in wet-avalanche flows are large, heavy and poorly sorted in comparison to granules in dry avalanches (Jomelli and Bertran, 2001; Bartelt and McArdell, 2009). Sintered particle agglomerates and levee constructions with steep vertical shear planes are found in wet snow avalanche deposits, indicating
that cohesive processes are an important element of wet snow avalanche rheology (Bartelt et al., 2012c, 2015).

To model wet snow avalanche flow we extend ideas first suggested by Voellmy (1955) and adopted in the Swiss guidelines on avalanche calculation (Salm et al., 1990; Salm, 1993). Voellmy proposed a frictional resistance $\mathbf{S}_\Phi = (S_{\Phi x}, S_{\Phi y})$ consisting of both a Coulomb friction $S_\mu$ (coefficient $\mu$) and
a velocity dependent stress $S_\xi$ (coefficient $\xi$):

$$\mathbf{S}_\Phi = \frac{\mathbf{u_\Phi}}{\|\mathbf{u_\Phi}\|} [S_\mu + S_\xi]. \tag{19}$$





Voellmy maintained that the Coulomb friction term decreased to zero $S_\mu \to 0$ for two extreme avalanche flow regimes: powder snow avalanches and wet snow avalanches. In these cases, avalanche velocity was determined purely by the velocity dependent stress $S_\xi$. This is given by

$$S_\xi = \rho_\Phi g \frac{\|\mathbf{u}_\Phi\|^2}{\xi}. \tag{20}$$

The Coulomb friction term was neglected for powder avalanche flow because of the dispersive, fluidized character of the avalanche core. In wet snow avalanche flow, the decrease of Coulomb shear stress is due to meltwater lubrication. To model the decrease in friction from either dispersion or meltwater lubrication we make the Coulomb stress dependant on the configurational energy $R_\Phi^V$ and meltwater water content $h_w$

$$S_\mu = \mu(R_\Phi^V, h_w) N_K \tag{21}$$

to arrive at a general friction law, valid for both dry and wet avalanche flows. This relationship will model the decrease in friction when the avalanche is highly fluidized and when the water content reaches a sufficient amount that lubrication cannot be neglected.

Because we employ a depth-averaged model to calculate the bulk avalanche temperature $T_\Phi$ we have no information to define the depth in the avalanche flow core where melting occurs. The dissipation rate $\dot{Q}_w$ depends on the internal shear distribution, which can be concentrated at the bottom surface of the avalanche, or distributed over the entire avalanche flow height. The spatial concentration of meltwater will therefore determine how the meltwater lubricates the flow. To account for the spatial distribution of meltwater in a depth-averaged model, we use the following two-parameter lubrication function to replace the standard Coulomb friction coefficient $\mu$

$$\mu(R_\Phi^V, h_w) = \mu_w + (\mu_d - \mu_w) \exp\left[-\frac{h_w}{h_s}\right]. \tag{22}$$

where $\mu_d$ is the dry Voellmy friction coefficient, $\mu_w$ is the limit value of lubricated friction (Voellmy assumed this value to be $\mu_w = 0$ in the limiting case) and $h_s$ is a scaling factor describing the height of the shear layer where meltwater is concentrated (Fig. 4). The dry friction $\mu_d$ depends on the avalanche configuration,

$$\mu_d = \mu_0 \exp\left[-\frac{R_\Phi^V}{R_0}\right] \tag{23}$$

where $\mu_0$ is the dry Coulomb friction associated with the flow of the co-volume, which we take to be $\mu_0 = 0.55$, see (Buser and Bartelt, 2015). The parameter $R_0$ defines the activation energy for fluidization, which is a function of the particle cohesion (Bartelt et al., 2015).

Meltwater production is considered as a constraint on the flow temperature of the avalanche: the mean flow temperature $T_\Phi$ can never exceed the melting temperature of ice $T_m = 273.15\ K$. The energy for the phase change is given by the latent heat $L$

$$\dot{Q}_w = L\dot{M}_w \tag{24}$$





under the thermal constraint such that within a time increment $\Delta t$

$$\int_0^{\Delta t} \dot{Q}_w dt = M_\Phi c(T_\Phi - T_m) \qquad \text{for} \qquad T > T_m. \tag{25}$$

Obviously, when the flow temperature of the avalanche does not exceed the melting temperature, no latent heat is produced, $\dot{Q}_w = 0$. The length of the time increment is defined by the numerical time integration scheme of the vector equations.

The mass of meltwater in the avalanche core $M_w$ is characterized by the height $h_w$ defined by the density of water $M_w = \rho_w h_w$. This height, measured from the avalanche running surface, is compared to the height $h_s$. We approximate the height $h_s$ using measured shear layers of wet avalanche flows which show $0.01 \text{ m} \leq h_s \leq 0.10 \text{ m}$, see Dent and Lang (1983); Dent et al. (1998); Bartelt et al. (2005); Kern et al. (2009). When the water content reaches the height $h_w$ compared to

the shear layer height $h_s$ the friction function $\mu(R_\Phi^V, h_w)$ decreases according to Eq.22 (see Fig. 4) The parameter $\mu_s$ defines the Coulomb friction when the layer $h_s$ is saturated, $h_w \approx h_s$. We take $\mu_w$ = 0.12. This ensures that dense, non-fluidized wet snow avalanches will continue to flow on slopes steeper $7°$ when they contain fully saturated lubrication layers.

### 2.4   Initial and boundary conditions

The Codelco Andina mine operates three automatic weather stations that measure air temperature, snow surface temperature, air pressure, wind speed, precipitation and incoming/reflected short wave radiation, see Table 1 and Fig. 1. The distance between the closest weather station and the avalanche paths varies between 0.5 km and almost 4.0 km. The meteorological data are used to run SNOW-PACK simulations (Bartelt et al., 2002; Lehning et al., 2002) that provide the snow temperature,

density and initial water content in the release zone ($T_0$, $\rho_0$, $\theta_0^w$) and snowcover ($T_\Sigma$, $\rho_\Sigma$, $\theta_\Sigma^w$). Snow pits are dug by the winter operation crew at regular intervals to supplement the measured/simulated snowcover data.

The release areas in the case studies are located between 3085 m and 3600 m; the weather station used here to drive the SNOWPACK simulations is located at 3520 m. The small elevation

difference between the release zones and the weather station provides sufficient accuracy in snow and meteorological data. However, surface energy fluxes are influenced by the slope exposition. To get representative simulations for potential avalanche release zones, virtual slope angles of $35°$ are used, shortwave radiation measured at the meteorological station as well as snowfall amounts are reprojected onto these slopes, taking into account slope angle and aspect (Lehning and Fierz, 2008).

Meteorological data from the winter operation building at the valley bottom (Lagunitas 2720 m, see Fig. 1) are also available. Thus, it was possible to estimate the precipitation and temperature gradients existing between the weather station location and the winter operation building and there-fore infer the snow cover conditions along the selected avalanche paths. To estimate the fracture and erosion depths for each case study we considered SNOWPACK simulations using Richards





equation for liquid water flow (Wever et al., 2014), which is able to reproduce the accumulation of liquid water at microstructural transitions inside the snowpack (Wever et al., 2015). Therefore we identify the interface below water accumulations as fracture points (Kattelmann, 1985; Mitterer et al., 2011; Takeuchi and Hirashima, 2013). Subsequently, the simulations provide fracture depth, average snow density, temperature and liquid water content of the slab, which extends from the

depth of the maximum liquid water content to the snow surface. The SNOWPACK estimations are validated with field measurements when the access are possible.

The initial avalanche release volume $V_0$ is calculated by estimating a release area $A_0$ and a mean fracture depth $h_0$. Point release avalanches are specified by defining a small triangular shaped release

area where the upper apex of the triangle is located at the release point. The triangular area together with the fracture height defines the initial release volume. The location of the release areas is based on observed releases for a particular track. This information has been collected and documented by the road safety crew.

The fracture $h_0$ heights and erosion layers $l_\Sigma$ are not specified automatically. The road manage-

ment crew studies the SNOWPACK results to identify layers where meltwater accumulates. This can be at the bottom of the snowpack, leading to full depth avalanche releases, or it can be at an interface between two snow layers. The mean snow temperature, density and moisture content of the release zone and erosion layers, however, are defined from the simulation data after the fracture and erosion depths have been defined. At present the procedure is not automatized to allow the

safety crew to explore different release and erosion scenarios.

## 3 Case studies

The "Cajón del rio Blanco" valley contains over 100 avalanche tracks. In the following we investigate five documented events that represent avalanche activity in the mine. The avalanches are

designated: **CCHN-3** Caleta Chica North, **CG-1** Cobalto, **LGW-2** Lagunitas West, **BN-1** Barriga North and **CV-1** Canaleta East (Table 1). The first four cases are spontaneous point release wet avalanches that released in periods of high temperature (isothermal snowcovers). These particular avalanches were selected because they reached the primary industrial road, endangering workers or interrupting mine logistics and communication. The avalanches were subsequently well documented

by the winter operation crew. The fifth avalanche also reached the road and was documented by an observation drone providing better runout, deposition and spreading data. This avalanche released as a slab and entrained moist, warm snow. In all five cases high-resolution digital elevation models, 2m resolution, of the terrain are available.

For the five case studies field measurements were carried out. The field measurements consisted



**Table 1.** Summary of five avalanche case studies. Parameters with the subscript '0' denote quantities related to the release mass. Parameters with the subscript '$\Sigma$' denote quantities related to the eroded mass. The entrainment $h_\Sigma + \Delta h_\Sigma$ denotes the amount of eroded snow with its respective decrease in eroded height per 100m of altitude. The value $\Delta\, SST_{12h}$ is the change in snow surface temperature in the last 12 hours before the avalanche released.

| Name | CCHN-3 | CG-1 | LGW-2 | BN-1 | CV-1 |
|---|---|---|---|---|---|
| Date | 14.08.2013 | 07.09.2013 | 09.09.2013 | 09.09.2013 | 19.10.2015 |
| Section | 3.1 | 3.2 | 3.3 | 3.4 | 3.5 |
| Figure | Fig. 6 | Fig. 7 | Fig. 8 | Fig. 9 | Fig. 10 |
| Measurements | GPS | GPS | GPS | GPS | Drone |
| Air temperature (°C) | 3.7 | -3.0 | 8.3 | 7.8 | -1.0 |
| New snow$_{72h}$ (m) | 0.0 | 0.4 | 0.28 | 0.28 | 0.0 |
| Snow surface temperature (°C) | -2.1 | -1.1 | -0.08 | -0.2 | -0.1 |
| $\Delta\, SST_{12h}$ (°C) | 16.1 | 11.8 | 5.2 | 9.2 | 2.0 |
| Parameter | **CCHN-3** | **CG-1** | **LGW-2** | **BN-1** | **CV-1** |
| $h_0$ Release depth (m) | 0.35 | 0.28 | 0.25 | 0.25 | 1.1 |
| $V_0$ Release volume (m$^3$) | 110 | 257 | 222 | 98 | 2477 |
| $\rho_0$ Release density (kg/m$^3$) | 250 | 300 | 355 | 349 | 272 |
| $\theta_0^w$ Water content (%) | 2.3 | 3.0 | 2.7 | 3.5 | 3.7 |
| $T_0$ Release temperature °C | -0.1 | -1.5 | -0.09 | -0.2 | -0.1 |
| $h_\Sigma$ Entrainment height (m) | 0.30 - $\Delta$0.05 | 0.40 - $\Delta$0.07 | 0.30 - $\Delta$0.05 | 0.40 - $\Delta$0.05 | 0.90 - $\Delta$0.05 |
| $\rho_\Sigma$ Entrainment density (kg/m$^3$) | 250 | 300 | 355 | 349 | 272 |
| $T_\Sigma$ Entrainment temperature °C | 0 | 0 | 0 | 0 | 0 |
| $\theta_\Sigma^w$ Volumetric water content (%) | 2.3 | 3.0 | 2.7 | 3.5 | 3.7 |
| $V_\Phi$ Deposition volume (m$^3$) | 3050 | 5150 | 10020 | 8770 | 8265 |
| $\rho_\Phi$ Deposition density (kg/m$^3$) | 450 | 450 | 450 | 450 | 450 |
| $V_\Phi/V_0$ Growth index | 28 | 20 | 45 | 90 | 4 |

of: GPS measurements (see Table 2) and manual measurements of the avalanche deposit heights along several transects perpendicular to the main flow direction (see Fig. 5). For the **BN-1** and **LGW-2** cases it was possible to reach the release area and measure the amount of snowcover eroded by the avalanche. Erosion measurements were conducted using a marked depth probe along the avalanche path (see Fig. 5 and Table 2). Due to the steep terrain and mine regulations those mea-

surements could not be performed for the **CCHN-3** and **CG-1** cases near the release areas. Erosion height measurements could only be carried out in and immediately above the main deposition area. For the **CV-1** avalanche aerial photography is available from a drone flight, (Fig. 5c).

     The measured meteorological data was used to drive the SNOWPACK simulations. Since the time of release of all avalanche events is known, the simulated snowcover data at the time of avalanche





**Table 2.** Summary of the GPS measurements by the Codelco Andina winter operation crew and the author. The measurements were taken with a GARMIN Etrex vista HCx device with an accuracy of ±2-5 m. Erosion depth measurements were taken at the erosion areas together with the GPS points, (see Fig. 5)

| Deposits Outline | | | | Erosion area | | | | | |
| BN-1 | | LGW-2 | | BN-1 | | | LGW-2 | | |
| Latitude | Longitude | Latitude | Longitude | Erosion depth | Latitude | Longitude | Erosion depth | Latitude | Longitude |
|---|---|---|---|---|---|---|---|---|---|
| -33.081576 | -70.250943 | -33.087515 | -70.258377 | Release point | -33.079659 | -70.248477 | Release point | -33.085986 | -70.262448 |
| -33.082093 | -70.258954 | -33.086527 | -70.258249 | 37 cm | -33.080922 | -70.249719 | 41 cm | -33.086028 | -70.261791 |
| -33.082246 | -70.252448 | -33.086833 | -70.257787 | 39 cm | -33.081240 | -70.249108 | 39 cm | -33.086351 | -70.261687 |
| -33.081867 | -70.252741 | -33.086350 | -70.256112 | 32 cm | -33.08437 | -70.250708 | 36 cm | -33.086338 | -70.260227 |
| -33.081493 | -70.252583 | -33.086765 | -70.255986 | 29 cm | -33.081902 | -70.250170 | 29 cm | -33.087102 | -70.259876 |
| | | -33.086911 | -70.255715 | | | | 33 cm | -33.086338 | -70.259062 |
| | | -33.087569 | -70.255689 | | | | 32 cm | -33.086443 | -70.258577 |
| | | -33.088329 | -70.256169 | | | | | | |

release was used to determine the input values. These values are reported in Table 1.

### 3.1   Caleta Chica North, CCHN-3

The **CCHN-3** is a long, narrow and steep avalanche path that starts at a ridge located at an elevation of 3685 m (Fig. 6). The path contains a steep gully that includes track segments with steep inclinations of more than 60 degrees. The avalanche path ends directly above the industrial road

at 2700 m. Although the gully is narrow, the avalanche collects enough snow to endanger the industrial road due to the long distance between the release zone and the deposition area.

On the $14^{th}$ of August 2013 around 17:30 a point release avalanche started at the top of the avalanche path reaching the industrial road with a final volume of 2500 m$^3$ (estimated by the winter

operation crew, see Fig. 9a and 3a). On the $12^{th}$ of August 0.15 m of new snow was measured at 3500 m. A 24 hour period of cloudy weather followed the snowfall. The $14^{th}$ of August was the first clear sky day after the snow fall from the $12^{th}$ of August. The air temperature at the estimated release time was 3.7 °C at 3550 m.

### 3.2   Cobalto, CG-1

The **CG-1** avalanche path is located 2 km to the north (see Fig. 1) of the **CCHN-3** track with similar west exposition. The track starts at 3465 m and ends at the industrial road at 2450 m (Fig. 7). The release is located at a steep inclination located below a ridge. The track is channelized between two vertical rock pillars. The gully between the pillars has an inclination between 60 and 70 degrees for the first 500 vertical meters of drop. The track becomes progressively flatter (about 40-45 degrees)

and wider. For the last 300 m of elevation drop the gully is between 50 to 70 meters wide and the avalanche can entrain large amounts of snow. The deposition area is located on a cone shaped debris fan above the industrial road (see Fig. 7). The surface of the debris fan contains large blocks.





On the $7^{th}$ of September, 2013 at 17:30 hours a point release avalanche started from the upper part of the gully, eroding the upper new snow layer. The avalanche reached the valley bottom stopping a few meters above the industrial road (see Fig. 7). The volume of the deposits was estimated to be approximately 7000 m$^3$. On the $6^{th}$ of September a 24 hour storm left 0.40 m new snow at 3500 m. At 2720 m the storm began as a rainfall, placing 7 mm of water in the snowcover. At higher elevations above 2720 m, the rain turned to snow depositing 0.10 m of moist new snow on the wet snowcover. At 2400 m only rain was measured. The winter operation crew made two snow profiles at the morning of the $7^{th}$ of August and estimated that the rain reached 2900 m, above this elevation all precipitation fell as snow.

### 3.3 Lagunitas West, LGW-2

The **LGW-2** avalanche path starts at 3250 m below a rock band and continues over an open slope with 40-45 degree inclination (Fig. 8). The track contains two five meters drops over rock bands before it gets progressively flatter, reaching an inclination of 30-35 degrees. The track finishes at 2800 m at the industrial road with a 25 degree inclination (Fig. 1).

At 14:30 hours on the $9^{th}$ of September, 2013 a point avalanche released below the upper rock band reaching a secondary industrial road. The $9^{th}$ of September was the first clear sky day after the three day storm and cloudy weather that started on the $6^{th}$ of September. The air temperature at the release time was 8.3 °C at 2720 m.

### 3.4 Barriga North, BN-1

The **BN-1** avalanche path starts directly in front of the winter operation building at 3100 m (Fig. 9). The release area has a southern exposition and is situated below a wide ridge with 40-45 degrees slope angle. Below the release zone, the avalanche path flattens and twists, the track becoming exposed to the west. The avalanche path ends on an industrial road at 2775 m.

At 17:30 hours on the $9^{th}$ of September 2013, three hours after the **LGW-2** release, a point avalanche released below the ridge. The avalanche eroded new snow in the flat area, passed the channel turn and reached the access road. The winter operation crew estimated the maximum avalanche deposits to be approximately 3.5 m in height; 2 m on average. The air temperature at the release time was 7.8 °C. The avalanche were observed by mine staff members. Low quality video recordings from mobile phones are available.



**Table 3.** Summary of input simulation parameters for the five calculation examples.

| Parameter | BN-1 | LGW-2 | CG-1 | CCHN-3 | CV-1 |
|---|---|---|---|---|---|
| Grid size (m) | 2 | 2 | 2 | 2 | 2 |
| $\mu_0$ (–) | 0.55 | 0.55 | 0.55 | 0.55 | 0.55 |
| $\mu_w$ (–) | 0.12 | 0.12 | 0.12 | 0.12 | 0.12 |
| $\xi_0$ (m s$^{-12}$) | 1300 | 1300 | 1300 | 1300 | 1300 |
| $\alpha$ (–) | 0.07 | 0.07 | 0.08 | 0.08 | 0.08 |
| $\beta$ (1/s) | 1.0 | 1.0 | 1.0 | 1.0 | 1.0 |
| $R_0$ (kJ/m$^3$) | 2 | 2 | 2 | 2 | 2 |
| $h_m$ (m) | 0.1 | 0.1 | 0.1 | 0.1 | 0.1 |
| $\kappa$ (–) | 1 | 1 | 1 | 1 | 1 |

### 3.5 Canaleta East, CV-1

The **CV-1** is a steep avalanche path that has two main sections (Fig. 10). The starting point is a 40 degrees steep rock band which accumulates snow transported by north westerly winds. Below the rock band appears a 20 meter high cliff that leads to a steep and narrow 50 meter long gully. The avalanche path finally opens onto a graveled 40-42 degrees steep fan. The fan is located directly above the industrial road.

On the 19$^{th}$ of October 2015 at 18:15 hours a wet slab released from the rock band 200 meters above the industrial road. The avalanche flowed over the cliff and then into the gully, eroding the remaining snow cover. The snow on the fan was also eroded. The avalanche stopped after crossing the industrial road leaving about 10000 m$^3$ of mass in the deposits. Between the 13-14$^{th}$ of October, 97 cm of new snow were measured at Lagunitas operations center (400 meters away from the avalanche path). After the snow fall between the 16$^{th}$ and 18$^{th}$ of October air temperatures between 6°C to 9°C were measured. In the last three hours before the release three millimeters of rain were measured in Lagunitas.

### 4 Simulation results

The primary goal of the case study simulations is to reproduce avalanche runout using the measured and simulated snowcover initial ($h_0$, $V_0$, $\rho_0$, $T_0$, $\theta_0^w$ ) and boundary ($h_\Sigma$, $V_\Sigma$, $\rho_\Sigma$, $T_\Sigma$, $\theta_\Sigma^w$) conditions, friction parameters were not allowed to vary from one case study to the next. The selected friction parameters are presented in Table 3. All simulations were performed on a 2m x 2m digital elevation model. The terrain model was obtained using 2m laser scanning measurements performed in 2011 and 2013. The calculation domains contained up to 25000 cells, but calculation times were less than 20 minutes on a standard PC.


### 4.1 Runout, flow width and deposition

Figs. 6, 7, 8, 9 and 10 depict the calculated maximum flow height and runout. Photographs of
the real events are provided in the figures to allow a direct comparison. In all five case studies the
avalanches start on steep slopes. Flow paths were all correctly modeled, including the location where
the avalanche cut the road. Calculated runout distances are in good agreement with the GPS mea-
surements made by the road operation teams. Three flow fingers developed in the **LGW-2** avalanche
were all reproduced by the model, Fig. 8. No channel break-outs were observed or calculated for the
channelized avalanches **CG-1** and **CCHN-3**. In both cases, the avalanches followed a steep, deep
and twisted channel. All calculations were made with the same model parameters with the exception
of the generate parameter $\alpha$, which depends on the avalanche track steepness and changing curvature
and twists. In the **BN-1** and **LGW-2** avalanches it was required to use a slightly lower production
value ($\alpha$) for the random kinetic energy, $\alpha = 0.07$ (in comparison to $\alpha = 0.08$ for the other case
studies), see Table 3.

Not only was it possible to reconstruct the avalanche runout, but also the avalanche flow width
(Fig. 11). For example, the measured width of the **BN-1** avalanche depositions on the road at 2750 m
elevation was 82 m; the calculated width was 90 m. The measured width of the **CV-1** avalanche was
132 m at 2720 m (drone measurements); the calculated width 139 m. That is, the model predicted
somewhat larger deposition widths indicating a slight spreading before stopping, especially for the
three open slope avalanches **BN-1**, **LGW-2** and **CV-1**. Fig 11 compares the observed maximum
deposition heights with the calculated deposition heights at the road. In the case study **CCHN-3** the
calculated deposition heights are lower than the maximum observed heights because the avalanche
ran over old 2 m high avalanche depositions, which are not included in the simulations. If the height
of the old deposits is added to the simulation results, a good agreement between calculated and
observed deposition heights is achieved.

### 4.2 Avalanche temperature and meltwater production

Calculated avalanche temperatures are shown in Fig. 12. In the five case studies the calculated
temperature of the flowing snow $T_\Phi$ reached the snow melting temperature $T_m = 0°$. This indicates
that frictional dissipation produced meltwater over considerable distances along the avalanche path,
for all five case studies. Avalanches that started with release temperature below $T_0 < 0°C$ (**CG-1**
and **CV-1**) quickly reached the melting temperature. Total meltwater produced, at a specific point
on the avalanche track, reached peak values of 3 mm m$^{-2}$. Once produced, meltwater is advected
with the speed of the avalanche, leading to regions in the flow where meltwater accumulates. Melt-
water accumulations can be as high as 60 mm m$^{-2}$, see Figs. 13 and 16. The advected meltwater
accumulations determine the value of Coulomb friction, see Fig. 13, which is a function of both the
configurational energy and the amount of meltwater.


### 4.3 Avalanche velocity and fluidization

Figure 18 depicts the maximum velocity calculations of the **BN-1** and **LGW-2** case studies. The
flow velocities of the avalanches did not exceed 15 m s$^{-1}$; the maximum calculated velocities in
the runout zone never exceed 10 m s$^{-1}$. Avalanche velocities could be roughly estimated using
the mobile phone video recordings. The velocity measurements (about 10 m s$^{-1}$) coincide with
these predictions. Unfortunately the recordings are not accurate enough to perform a more precise
analysis.

For such steep terrain, higher velocities are to be expected. However, the avalanches did not
fluidize completely. The avalanches remained in a frictional flow regime with relatively high flow
densities, $\rho_\Phi \approx 300$ kg m$^{-3}$, see Fig. 16. At the point of maximum flow velocity (15 m s$^{-1}$),
the **BN-1** avalanche had a minimum flow density of $\rho_\Phi = 305$ kg m$^{-3}$. Similarly, at the point of
maximum flow velocity (18 m s$^{-1}$), the **LGW-2** avalanche had a minimum flow density of $\rho_\Phi =$
302 kg m$^{-3}$. In the runout zone the minimum flow densities were on the order of $\rho_\Phi = 450$-480 kg
m$^{-3}$. This value is very close to the final deposition density of $\rho_\Phi = 500$ kg m$^{-3}$ The maximum
configurational energies reached 80-100 kJ/m$^2$, see Fig. 13.

### 4.4 Entrainment

The numerical results underscore the important role of snow entrainment. The increase in avalanche
volume from release to deposition for four case studies is depicted in Fig. 14. The initial release
volumes $V_0$ are defined at $t = 0$. For all point release case studies the initial volume $V_0 < 300$ m$^3$.
The final calculated deposition volumes $V_\Phi$ are $V_\Phi \approx 8700$ m$^3$ for the **BN-1** and $V_\Phi \approx 10000$ m$^3$ for
the **LGW-2** case studies. In the remaining two examples **CCHN-3** and **CG-1** the avalanches did not
entrain snow after the track midpoint. In these two examples there was no snow cover below 2900
m (see Figs. 6 and 7) . The growth indices for these avalanches are smaller, but nonetheless large.
The calculated growth indices (Fig. 14b) reach values between $V_\Phi/V_0 \approx 20$ and 90 indicating that
entrainment processes are controlling the avalanche size.

The two case studies with entrainment measurements (**BN-1** and **LGW-2**) are particularly impor-
tant. Dividing the calculated deposition volumes by the area measured by the winter operation crew
(see Fig. 5b) we found $h_\Phi \approx 2.4$ m deposit height in the **BN-1** case study and $h_\Phi \approx 1.3$ m in the
**LGW-2** case study. These results roughly agree with the field volume measurements, $h_\Phi \approx 3$ m and
$h_\Phi \approx 2$ m, respectively.

### 5 Discussion

The simulation results rely on accurate initial conditions (release volume, location and snow temper-
ature, density and liquid water content) and boundary conditions (track roughness, snowcover depth,
snow density, temperature and liquid water content) and not in changing the model parameters for



wet snow (which we kept constant). The model predicts dense flows with high flow density, congruent with observations of wet snow avalanche motion. Fluidization can occur in steep and rough terrain; however, runout is controlled by meltwater lubrication and therefore the changing material

properties of snow as it becomes warmer and wetter. This implies that snowcover conditions temperature, density and moisture content, which control the hydrothermal state of the flowing snow, must be included in the model formulation.

As the SNOWPACK simulations predicts isothermal snowcover at $T_\Sigma = 0°$ for the snow depth affected by the avalanches, the entrained snow temperature was set to zero degrees in all five case

studies, see Table 1. This approach could not be followed with the modeled snowcover water content which has no limiting value in an isothermal snowcover. Although SNOWPACK was used to predict snow water content (Wever et al., 2014), it was difficult to measure and validate the distribution of snow water content at lower altitudes and different expositions. For example, in the case **CG-1** the snowfall was preceded by rain making it difficult to calculate the snow water content which depends

on the variability of the rainfall.

The positions of all release zones were obtained from the eyewitness reports and post-event surveys. Entrainment depths for the simulations were also obtained from field measurements and event documentation. In the examples **LGW-2** and **BN-1** the erosion depths where measured along the path in several points (Fig. 5). Because the avalanches disrupted road traffic, the winter operation

crew could estimate deposition depths allowing good estimates of avalanche mass balance. The temperature, snow density and water content of the eroded mass are the key input information to predict accurate avalanche deposition volumes and runout distances. In the case of point releases, the release mass does not play an important role (Fig. 14) apart from defining the location of release and the triggering of the whole subsequent process.

The five examples contain mountain rock faces with well defined flow channels (**CG-1**, **CCHN-3**) as well as open slopes (**BN-1** and **LGW-2**) or a mix of them, **CV-1**. At release the avalanche mass spreads depending on the terrain features. In two of the five case studies, avalanche spreading is inhibited by the steep sidewalls of mountain gullies, a function of the topographic properties of the mountain. The remaining three examples are open slopes where the spreading angle is larger. The

spreading angle was accurately reproduced in all three case studies. Small avalanches are extremely sensitive to small topographic features therefore high resolution digital elevation models that accurately represent mountain ravines and channels are thus necessary to apply more detailed avalanche dynamics models to simulate small avalanches (Bühler et al., 2011).

The avalanche model simulates both fluidization and lubrication processes. This requires intro-

545 ducing depth-averaged equations for thermal energy (Vera et al., 2015), mechanical free energy (Buser and Bartelt, 2015) and meltwater (Vera et al., 2015). The degree of fluidization characterizes the avalanche flow regime: dry snow avalanches being associated with more fluidized, less dense flows (mixed flowing/powder avalanches) and wet avalanches being associated with less fluidized,




dense flows. The degree of fluidization is controlled by parameters ($\alpha$ and $\beta$) governing the production and decay of free mechanical energy $R$ (Buser and Bartelt, 2015). The production parameter $\alpha$ is made dependent on terrain roughness and is independent of the avalanche temperature and moisture content, in this work the values used correspond to the 7-8% (see Table 3) of the work done by the friction at the bottom surface. Highly plastic, wet particle interactions quickly dissipate any free mechanical energy leading to dense flows that can only fluidize in steep, rough slopes. We model this process by increasing the dissipation parameter $\beta$ to 1.0 for warm ,wet avalanches, (Buser and Bartelt, 2015). This produces dense flows in the frictional flow regime. In the four case studies the flow density in the runout zone is close to the deposition density $\rho_\Phi = 450$ kg m$^{-3}$, whereas in the steep track sections the flow density is somewhat lower $\rho_\Phi = 300$ kg m$^{-3}$ (see Fig. 16). Important is that the same model formulation is used for both dry and wet avalanches and fluidization is controlled by a combination of terrain (production of free mechanical energy) and wet snow granule properties, (dissipation of free mechanical energy). An important model assumption is that entrainment of moist wet snow is a completely dissipative process which does not introduce additional free mechanical energy into the avalanche core.

Therefore, our results indicate that fluidization cannot be responsible for long runout distances of wet avalanches. Snow chute experiments with wet snow, showing that cohesive interactions in the avalanche core further hinder fluidization (Bartelt et al., 2015), provide more evidence that wet snow avalanche mobility is strongly linked to the temperature and moisture dependent mechanical properties of wet snow (Voytokskiy, 1977). To investigate this hypothesis, we postulate that temperature and lubrication effects lead to a significant reduction of the Coulomb part of the Voellmy friction. A two parameter empirical relation between water content and friction $\mu$ was devised. A problem with depth-averaged models is that the distribution of meltwater in the avalanche height cannot be predicted from depth-averaged calculations of avalanche flow temperature, which depends on the slope perpendicular shear profile in the avalanche core. We assume that meltwater is concentrated in a shear layer whose height is in the order of magnitude of $h_m$. When this layer becomes saturated with meltwater, Coulomb friction is reduced to a sliding value of $\mu_s$, which we take, for now, to be constant $\mu_s = 0.12$. This value was selected based on our observations of wet snow avalanche flowing in slopes not flatter than 9°, (tan 9°= 0.12). The layer height was set to $h_m$= 0.1 m, indicating that shearing in wet avalanche flows is concentrated in a basal layer, (see Fig. 4). This is in agreement with velocity profile measurements of wet avalanche flows (Dent et al., 1998; Kern et al., 2009). The snow water content values obtained in the simulation results varied between 10-50 mm m$^{-2}$. Spreading such amount of water within the shear layer ($\approx h_m$) leads to water concentrations volume higher than 15% of volume water content. With such water concentration this avalanche layer is above the so-called capillary regime (Mitarai and Nori, 2006) where the interstitial water pressure is higher than air pressure and therefore lubrication occurs. Spreading the same amount of water content obtained in the model in a hypothetical larger shear




layer ($h_m \approx 1$m) leads to a lower water concentration and therefore to a higher $\mu$ which prevents the avalanche to reach the measured run out (see Fig. 17).

The model calculates the depth-averaged flow temperature from initiation to runout. In the five case studies the avalanche reached the melting point of snow-ice immediately after release due to the warm initial conditions. The entrainment of warm, moist snow enhanced the lubrication process. This is shown in Fig. 15. We made two calculations for the **LGW-2**, **BN-1** and **CCHN-3** avalanches. In the first calculation we set the release temperature to $T_0$ = -10°C and in the second calculation we used $T_0$ = 0°C, near the measured values. The difference in runout is large. The decrease of

Coulomb friction due to lubrication effects was essential for the point release avalanches to develop into long-running wet snow avalanches. For practical applications it is important that lubrication processes due to the (1) initial snow water content, (2) snow melting by frictional dissipation and (3) heat energy of entrained snow must all be taken into account.

    The method used to simulate the avalanche point release requires defining a small triangular

area. The ratio between the eroded snow volume and the initial snow volume is between 20 to 90 for the four point releases we studied in this paper. The initial area used to simulate the avalanche release does not affect the final run-out, velocity and avalanche deposit calculations. The model results emphasize that complete information of the snow cover is necessary to achieve accurate representations of the events. The model is sensible to variations in the initial snow cover

conditions, temperature and water content. For example, when colder snow is specified at release, the simulated avalanches stop immediately after release and do not reach the valley bottom. Given accurate initial conditions the model was able to back calculate runout distances, flow outlines and avalanche volumes. Therefore, with this model formulation, it is only possible to obtain realistic runout predictions with accurate snow cover data.


## 6   Conclusions

For mining companies road closure is associated with severe financial costs and winter operation crews must deliver runout warnings based on daily, perhaps hourly, meteorological information. Many existing avalanche dynamics models widely used in practice, (e.g. Christen et al. (2010);

Sampl et al. (2004); Sheridan et al. (2005); Mergili et al. (2012)), do not include the role thermal temperature, fluidization or snow liquid water content in their mechanical description of avalanche motion. As such, wide ranging flow parameters are required to model avalanche runout and velocity. These models therefore cannot be applied to forecast how avalanche activity will disrupt mining operations because they cannot take into account current measured and observed snow conditions.

To address this problem we developed a depth-averaged avalanche dynamics model that separates



the properties of flowing snow from the specification of initial and boundary conditions, which can be supplied by winter operation crews using a combination of weather stations and snowcover modeling. The avalanche model requires input parameters for fracture depth, snow temperature, snow density and water content in the release area and along the avalanche path. The meteorological

data provided by the automatic weather stations is representative at the altitude where the weather stations are located. However, the difference in altitude and exposition of the five different cases studies requires a method to extrapolate temperature, snowcover depth and liquid water content from the point locations of the automatic weather stations to the entire slope. For this purpose we applied the SNOWPACK model on virtual slopes matching the expositions with the studied slopes.

When it was possible to enter the slopes we used traditional snow profiles measurements performed by the winter operation crew to validate the SNOWPACK model predictions for temperature, density and water content.

Avalanche dynamics models have been traditionally applied to simulate large, dry, slab release

avalanches. The starting volumes of such avalanches are typically larger than $V_0 > 50,000$ m$^3$. The primary application is to prepare avalanche hazard maps which are based on extreme events with long return periods or to determine input parameters for the design of engineering structures. In this paper avalanche release mass was modeled using small triangular shaped release zones containing less than $V_0 \approx 100$ m$^3$ of snow. The application of an avalanche dynamics model to simulate

small, point release avalanches is novel and poses many new challenges. Five preconditions for the simulation of such small avalanche events are:

1. The availability of high resolution digital terrain models

2. Information concerning the location of the release zone

3. Simulation of snow entrainment to model avalanche growth

4. Reliable snowcover information, including snow density,temperature and liquid water content

5. Reliable parameter values linking mechanical properties to snow temperature (e.g. dissipation, dry and wet Coulomb friction $\mu_0$ and $\mu_w$, etc.)

This information is seldom available in its entirety. Although we can imagine the development of tools linking release zone delineation, snowcover modeling with avalanche dynamics simulations

in the near future, their application will remain restricted to regions of similar climate and terrain where they can be thoroughly tested and applied by expert users. The application of this system was tested for two winter seasons in the Andina mine (Chile). The encouraging results motivated us to test the operational application. Simulations coupled with accurate and continually updated snow cover and meteorological information is required to predict avalanche run outs and deposition

volumes. The model does not provide any indication whether the avalanche is going to release or



not, but if the avalanche releases the model gives a good indication of the potential run out distances and deposition volumes.

Finally, a primary goal of this work is to develop a model that allows small and frequent events
to be analyzed by comparison model computations to field measurements. It is no longer necessary to wait for rare and extreme events as the model parameters are defined as material constants which depend not on avalanche size, but snow temperature and moisture content. As more data can be obtained from field observations it should be possible to further refine the constitutive formulations for meltwater lubrication and snowcover entrainment. We have proposed simple relations for obviously
complex processes that clearly need further testing. Alternative formulations are possible. More small, frequent avalanches should be studied and documented for this purpose.

*Acknowledgements.* Financial support for this project was provided by Codelco Mining, Andina Division (Chile). We thank all Codelco avalanche alert center members: L. Gallardo, M. Didier and P. Cerda, together with the 'Mountain Safety' workers not only for their support, but also for their confidence, patience and
enormous help during the last four winters in the Andina mine.



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



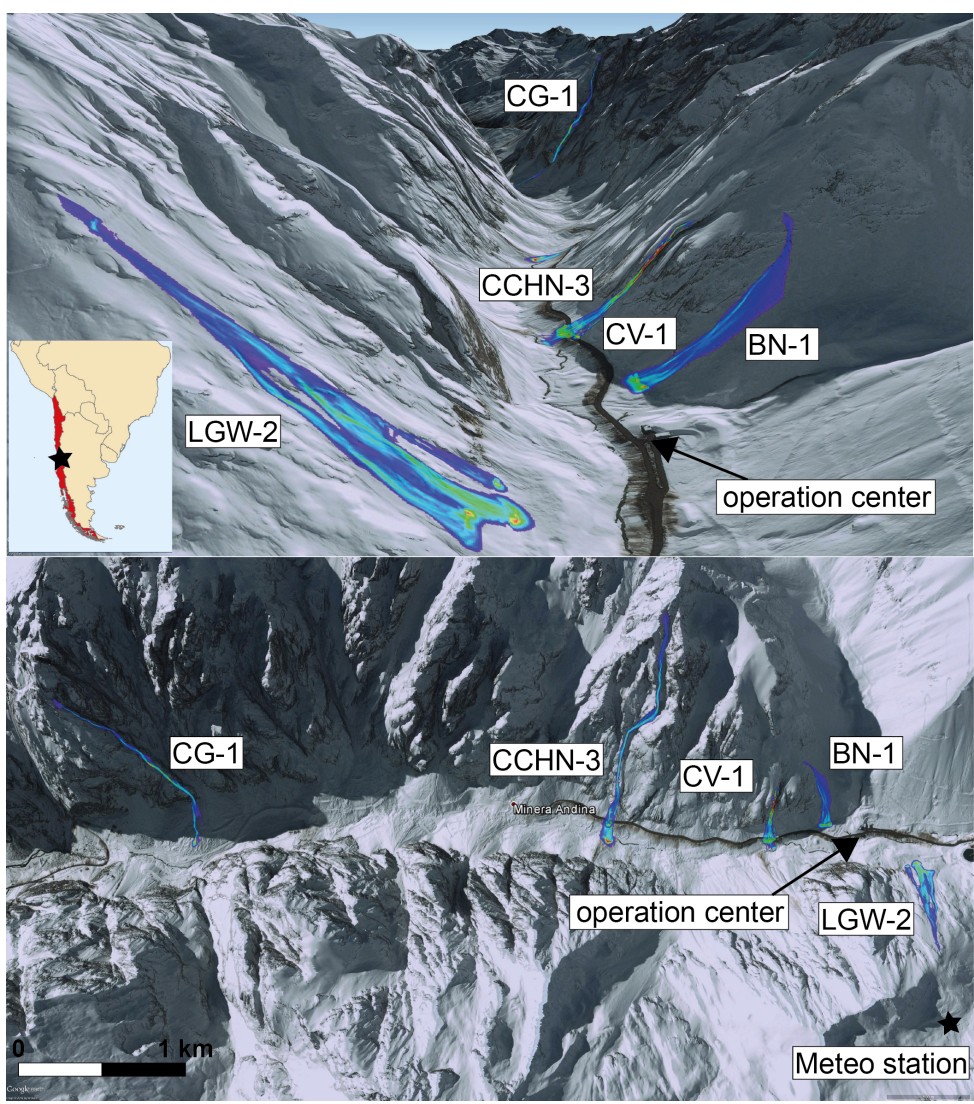

**Fig. 1.** Two and three-dimensional visualizations of a segment of the 35 km long mining road located in the
"Cajon del Rio Blanco" valley in the central Andes, Chile. The figure depicts the location of the five avalanche
(**CCHN-3** Caleta Chica North, **CG-1** Cobalto, **LGW-2** Lagunitas West, **BN-1** Barriga North and **CV-1** Canaleta
East) tracks in relation to the road and the location of two weather stations used to drive the SNOWPACK model.
One weather station is located at the 'Lagunitas' operation center at the valley bottom (2700 m). The automatic
weather station is located at an elevation of 3520 m. Picture obtained from Google Earth Pro.





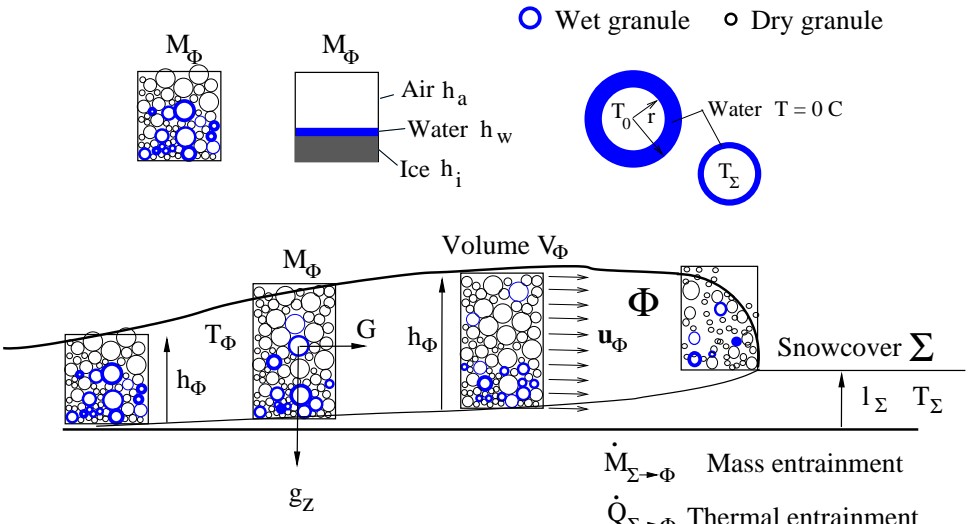

**Fig. 2.** The wet snow avalanche model divides the core $\Phi$ into flow columns with mass $M_\Phi$ and density $\rho_\Phi$. The mass is in the form of snow clumps and particles. The mean temperature of the mass $M_\Phi$ is $T_\Phi$. The densest packing defines the co-volume density $\rho_\Phi^0$. The particles can contain water, which we distribute on the surface of the particles where collisional and rubbing interactions takes place. The temperature of the granule interiors might be different from the surface temperature. The total mass of meltwater is $M_w$. This mass is bonded to the particles and travels with the avalanche. The avalanche model entrains snow mass $\dot{M}_{\Sigma \to \Phi}$ and thermal energy $\dot{Q}_{\Sigma \to \Phi}$. The avalanche is moving with speed $\mathbf{u}_\Phi$ in the slope parallel direction. Figure adapted from Buser and Bartelt (2015)

.




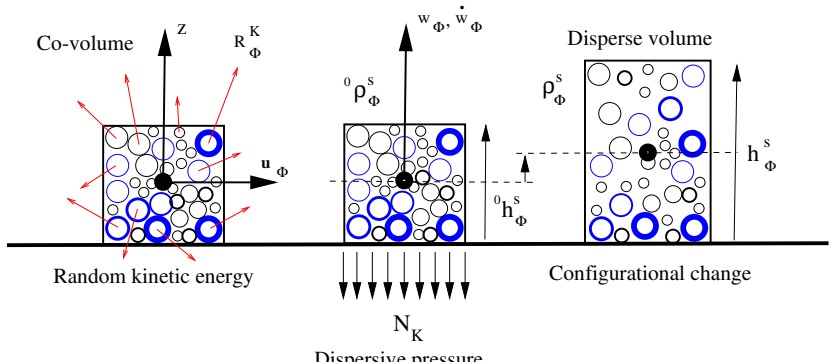

**Fig. 3.** A particle column in the avalanche. The densest particle packing defines the co-volume. When random kinetic energy is produced, not all the particles move in the same slope parallel velocity $\mathbf{u}_\Phi$. Particle interactions at the base serve to lift the particle column, producing a dispersive pressure $N_K$ and raising the center-of-mass of the column.

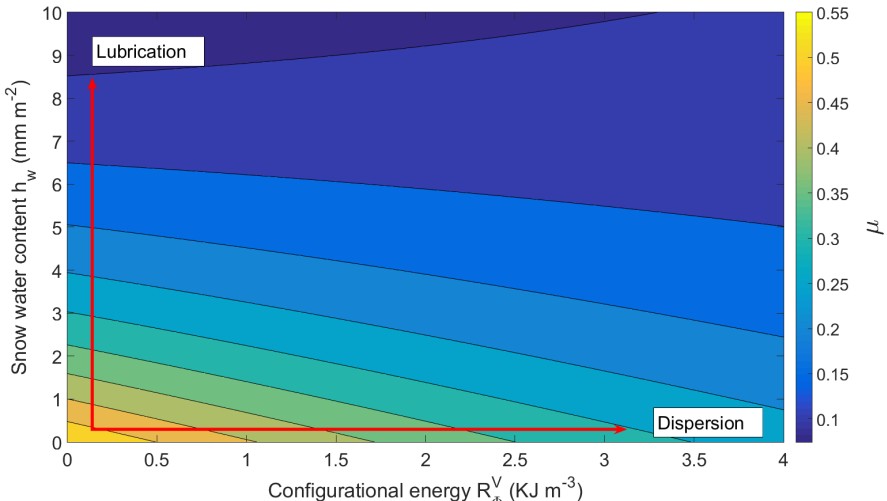

**Fig. 4.** Voellmy plot showing the dependancy of the friction parameter $\mu$ with configurational energy $R_\Phi^V$ and water content $h_w$ according to Eq.(22), $S_\mu \to 0$. Non-fluidized wet snow avalanches will not stop on slopes steeper than $9°$ when they contain fully saturated lubrication layers, $\mu(R_\Phi^V, h_w) \approx 0.15$ for $h_w = h_m$ and $R_\Phi^V \approx 0$.



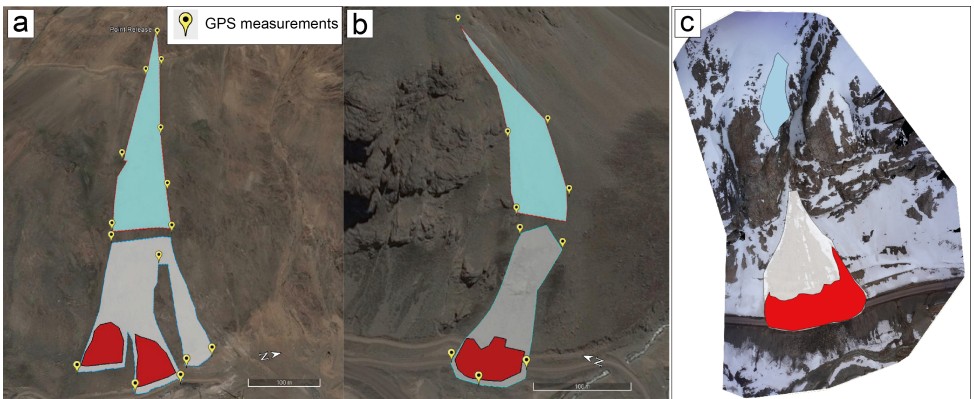

**Fig. 5.** Erosion-deposition measurements in the **LGW-2** (a), **BN-1** (b) avalanches and (c) **CV-1**. The yellow dots in (a) and (b) correspond to GPS measurements, see Table 2. For the **CV-1** avalanche (c) the erosion-deposition area was determined by a drone flight. The blue polygons show the erosion areas. The white polygons show the area where the avalanche was still eroding and already depositing mass (less than 1 meter deposits height). The red polygons inside the white polygon show the main deposit areas where the accumulations where higher than 1 meter. The measured deposit areas (red) were 7935 m$^2$ for **LGW-2**, 3726 m$^2$ for **BN-1** and 7373 m$^2$ for **CV-1**.



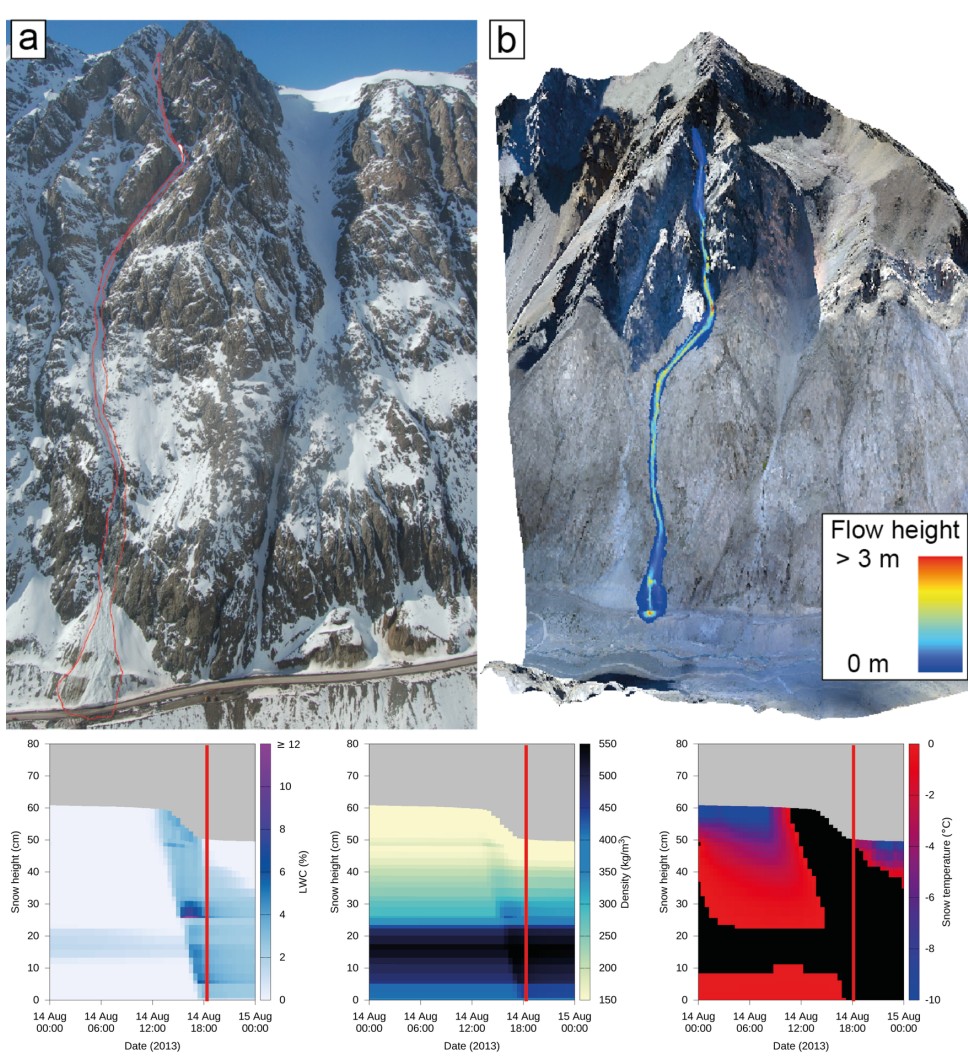

**Fig. 6.** (a) **CCHN-3** avalanche picture taken from the helicopter the day after the release. The point release was on the top of the steep gully on a rock face. The avalanche crossed the industrial road. (b) Calculated maximum flow height. The model correctly estimated the runout distance and the height of the avalanche deposits. Lower panel depicts the results of the SNOWPACK simulations, liquid water content, density and temperature, black color at the temperature plot denotes snow at 0°C with liquid water content greater than zero. The red line denotes the time of release.



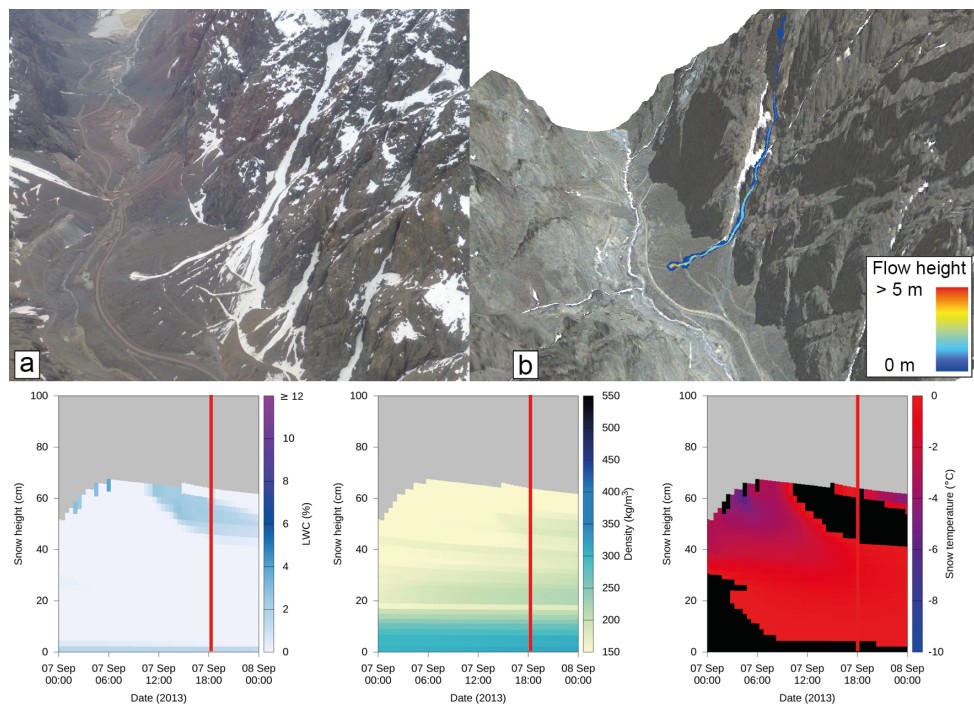

**Fig. 7.** (a) Avalanche path **CG-1**. Image taken from the helicopter the day after the release. The avalanche started at 3465 m but stopped eroding snow at 2900 m. The avalanche reached the valley bottom flowing over a scree surface. (b) Calculated maximum flow height. The model predicts the observed runout distance, avalanche outline and deposition volume. Lower panel depicts the results of the SNOWPACK simulations: liquid water content, density and temperature, black color at the temperature plot denotes snow at 0°C with liquid water content greater than zero. The red line denotes the time of release.



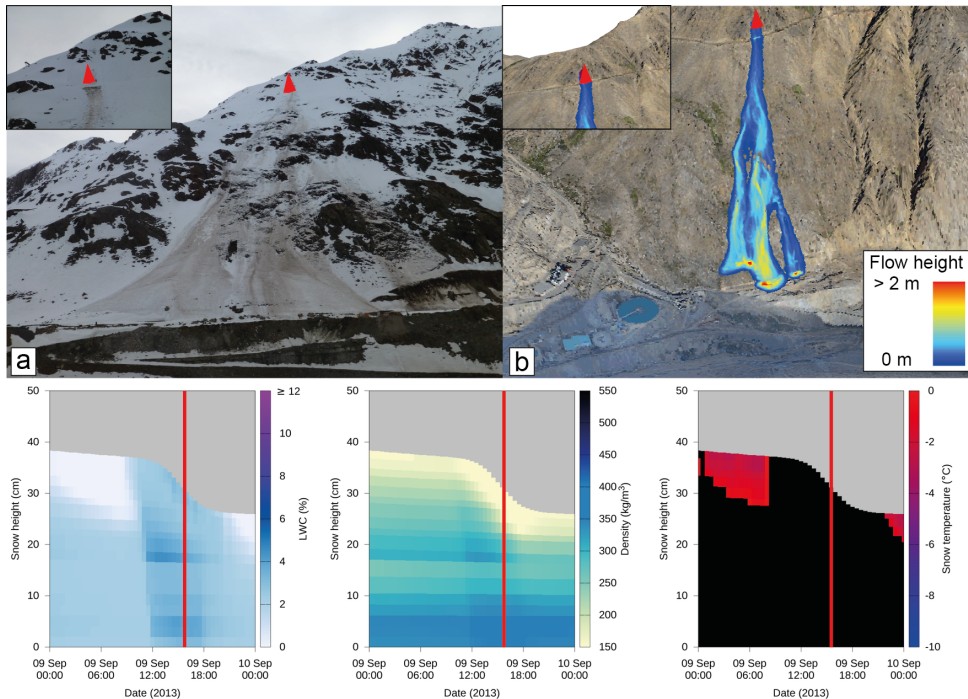

**Fig. 8.** (a) Avalanche **LGW-2** picture taken from the valley bottom. The avalanche released below a rock band and spread over the slope flowing over two rock bands before reaching a secondary road at the valley bottom. Top left shows a closer view from the release point. (b) Calculated maximum flow heights. The model correctly predicted the formation of three avalanche arms and therefore an accurate modelling of the avalanche outline. On the top left a closer view with the calculated release area (in red) is shown. Lower panel depicts the results of the SNOWPACK simulations: liquid water content, density and temperature, black color at the temperature plot denotes snow at 0°C with liquid water content greater than zero. The red line denotes the time of release.



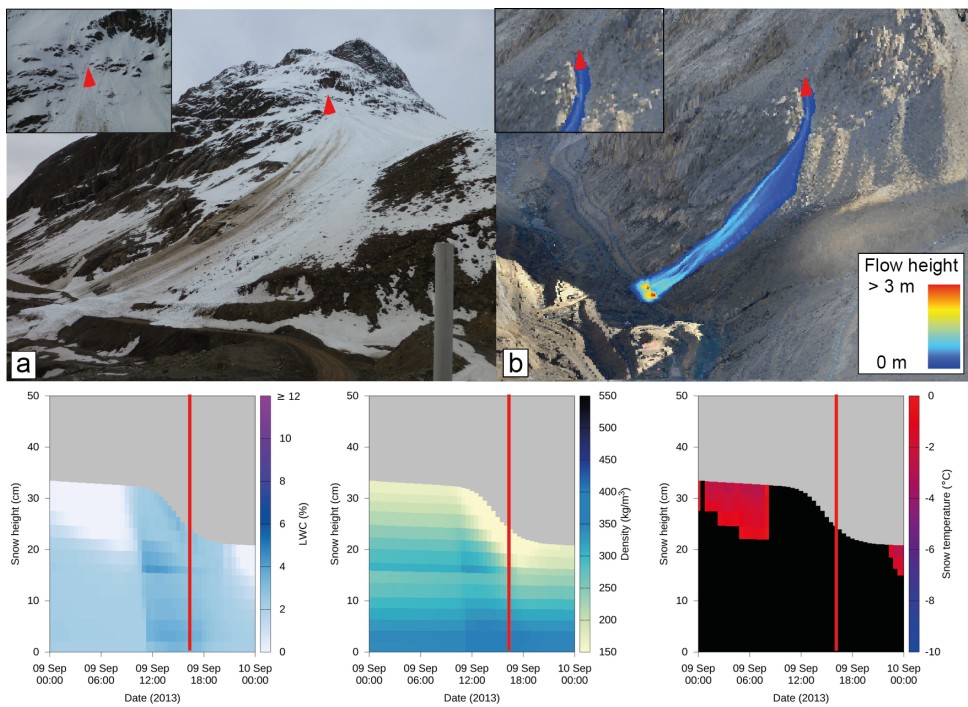

**Fig. 9.** (a) Picture of the **BN-1** avalanche taken from the Lagunitas operation center some minutes after the event. The avalanche crossed the road depositing on average 2 m of snow on the road. The top left inset provides a closer view of the point release. (b) Calculated maximum flow heights. The model accurately the avalanche spreading angle including the change in trajectory half way down the avalanche path. On the top left the calculated release area is shown in red. Lower panel depicts the results of the SNOWPACK simulations: liquid water content, density and temperature, black color at the temperature plot denotes snow at 0°C with liquid water content greater than zero. The red line denotes the time of release.


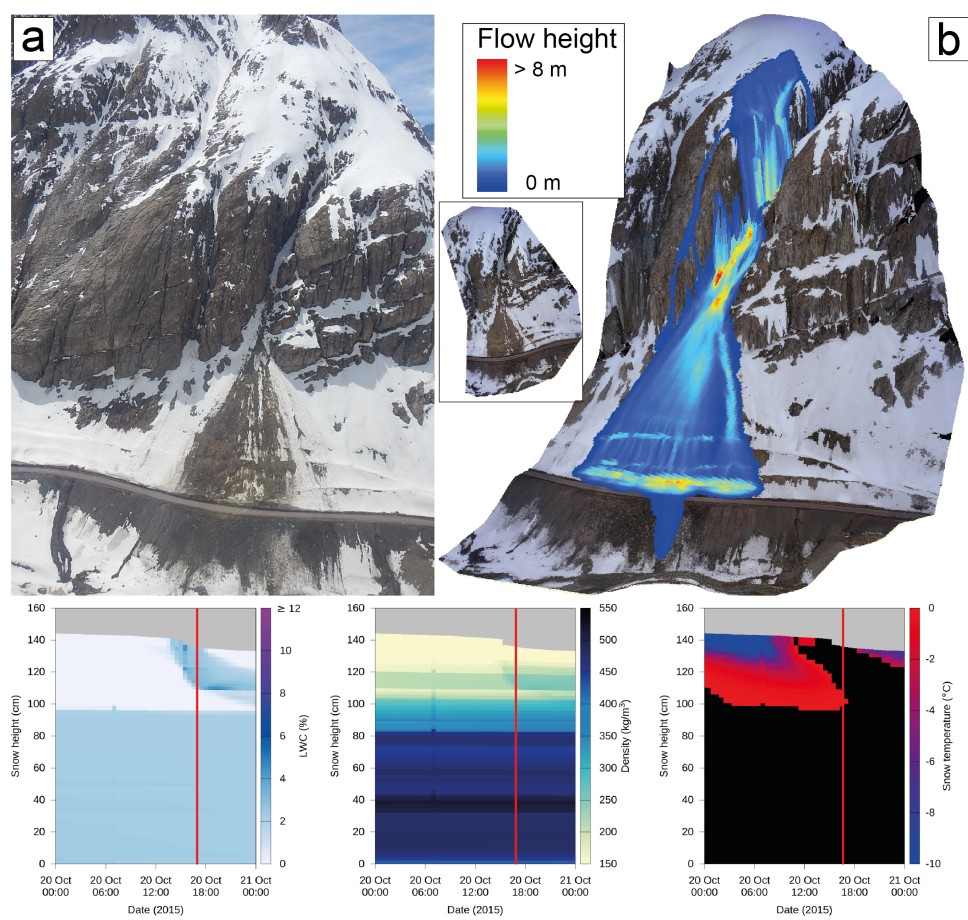

**Fig. 10.** (a) Picture of the **CV-1** avalanche taken from helicopter after the release. The slab was on the top the steep gully on a rock face. The avalanche crossed the industrial road leaving up to six meters of snow on the road. The avalanche deposits area and release area were photographed by a drone three days after the avalanche occurred (inset). Lower panel depicts the results of the SNOWPACK simulations: liquid water content, density and temperature, black color at the temperature plot denotes snow at $0°C$ with liquid water content greater than zero. The red line denotes the time of release.



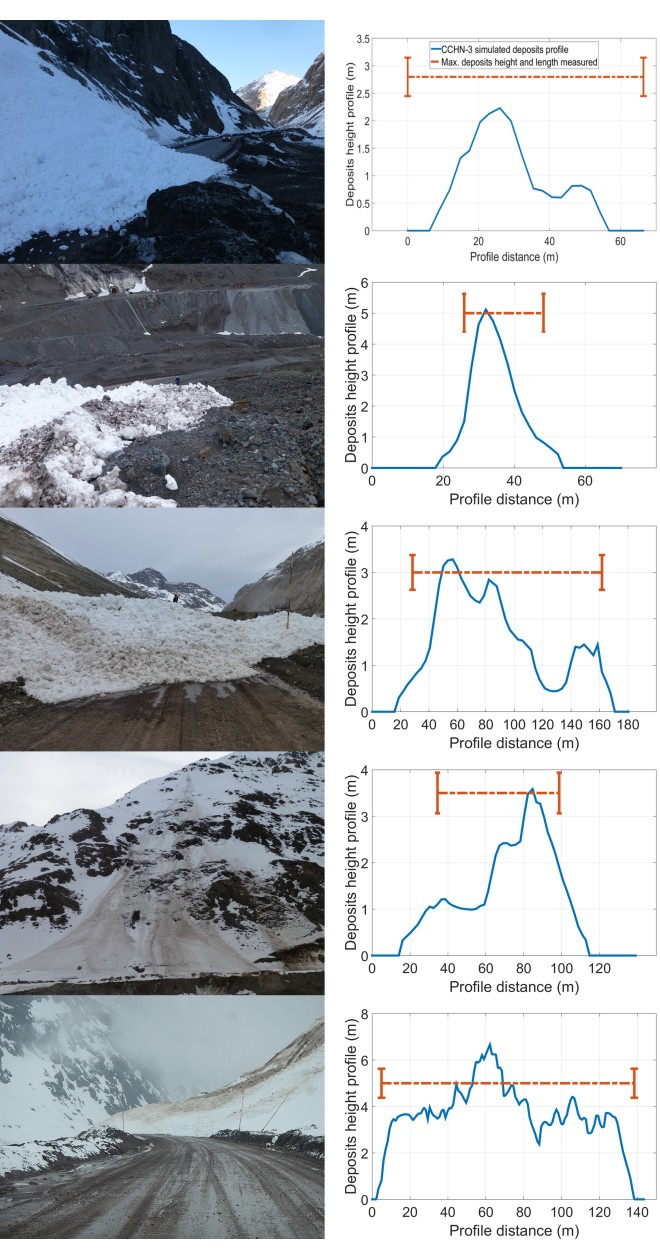

**Fig. 11.** Observed (left column) and calculated avalanche deposits (right column) on the road: (a) **CCHN-3**, (b) **CG-1** (c) **LGW-2** (d) **BN-1** and (e) **CV-1**. The outline and maximum height of the deposits were measured by the winter operation crew with a hand-held GPS device. The red lines in the plots depict the observed width and maximum height of the avalanche deposits.


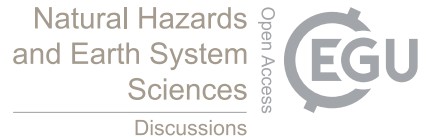

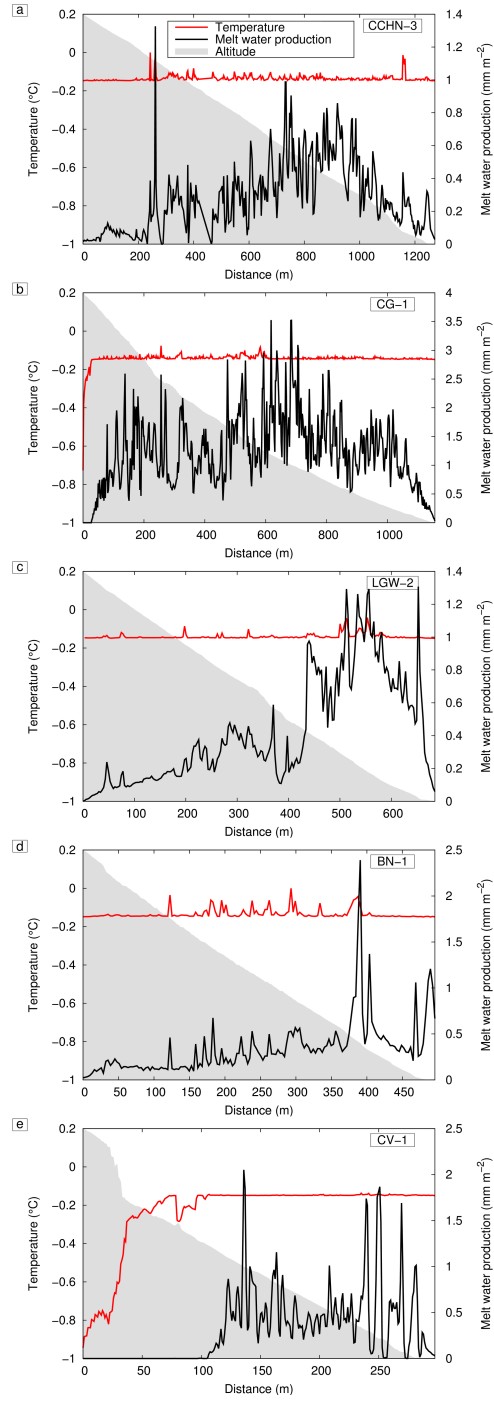

**Fig. 12.** Calculated temperature (red) and meltwater production (black). (a) **CCHN-3**, (b) **CG-1** (c) **LGW-2** (d) **BN-1** and (e) **CV-1**.The avalanche temperatures are close to $T_\Phi = 0°C$ from initiation to release. Frictional dissipation therefore led to an quick production of meltwater. The model predicted up to 3 mm m$^{-2}$ of meltwater. The grey shadow in the background indicates the elevation profile along the avalanche track.




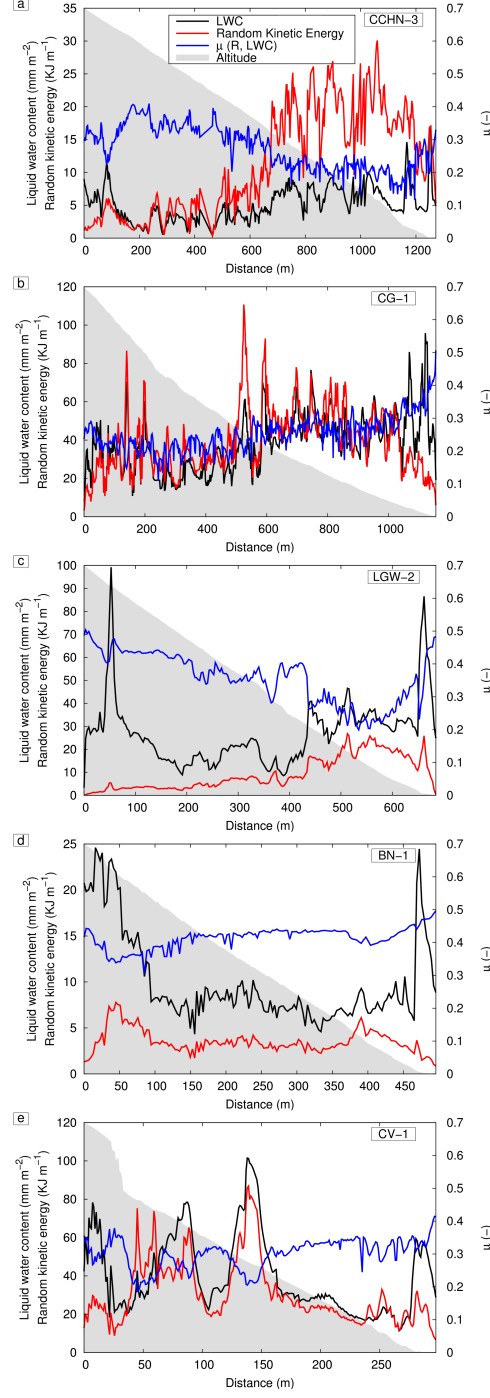

**Fig. 13.** Friction coefficient $\mu$ (blue), total liquid water content LWC (black) and total random kinetic energy $R$ (red): (a) **CCHN-3**, (b) **CG-1** (c) **LGW-2** (d) **BN-1** and (e) **CV-1**. Friction $\mu$ decreases with increasing LWC and random kinetic energy $R$. The grey shadow in the background indicates the elevation profile along the avalanche track.




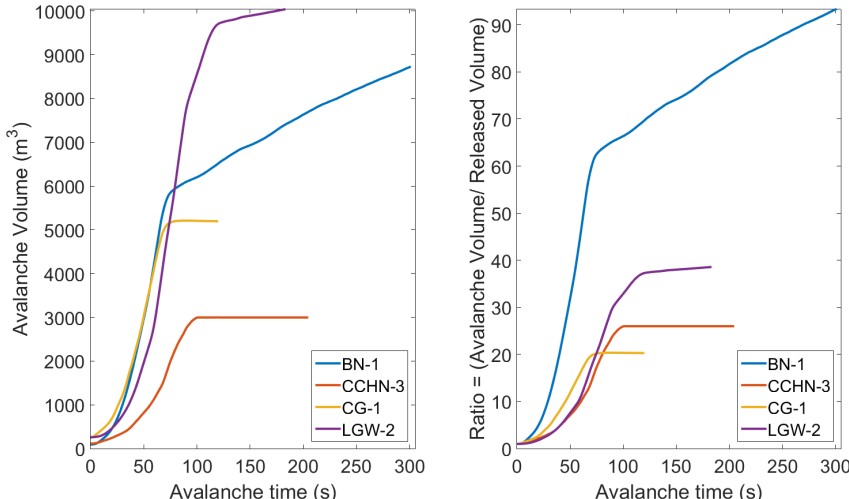

**Fig. 14.** (a) Ratio between the avalanche flow volume $V_\Phi$ and the initial release volume $V_0$ over time. In four of five case studies the ratio between the final volume and the initial simulated released volume is between $20 \leq V_0/V_\Phi \leq 90$. (b) Avalanche growth index. Flat curves indicate the time when the avalanches stopped entraining snow, (cases **CG-1** and **CCHN-3**).

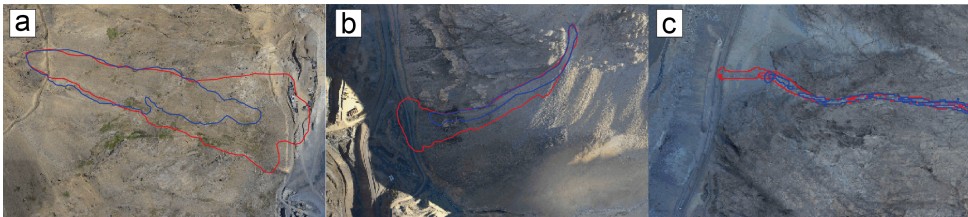

**Fig. 15.** Comparison between avalanche run out distance using cold ($T_0$ = -10°C, blue line) and warm snow ($T_0$ = 0°C, red line) for the (a) **LGW-2**, (b) **BN-1** and (c) **CCHN-3** case studies. Warm snow leads to more frictional melting and longer avalanche runout.





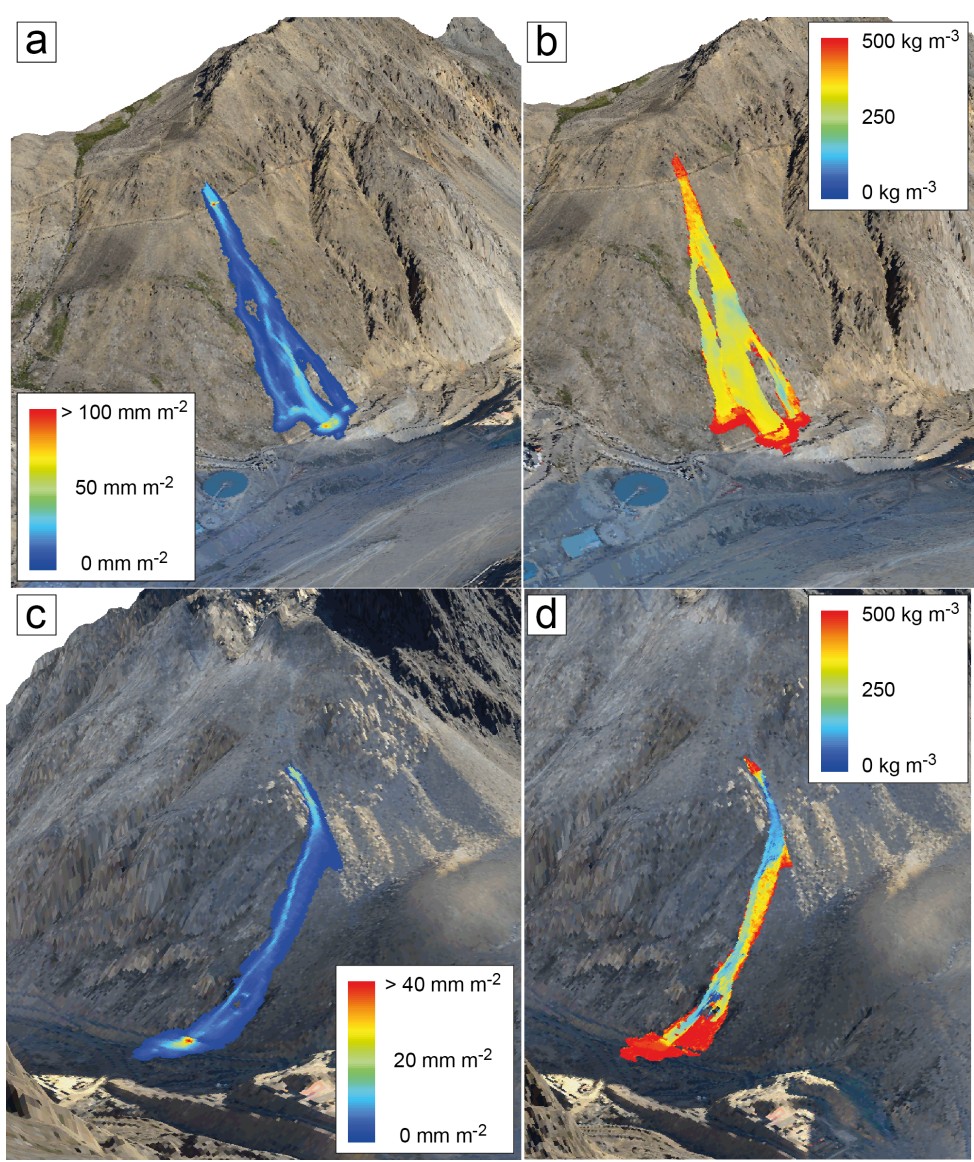

**Fig. 16.** Total calculated meltwater and flow density for the **LGW-2** and **BN-1** avalanches. (a) Total meltwater in **LGW-2** avalanche. (b) Flow density **LGW-2**. (c) Total meltwater in **BN-1** avalanche. (d) Flow density **BN-1**. In steep track sections the avalanche fluidized slightly (flow density $\rho_\Phi = 350$ kg/m$^3$). In the runout zones the avalanche densified. Deposition densities are $\rho_\Phi = 500$ kg/m$^3$.





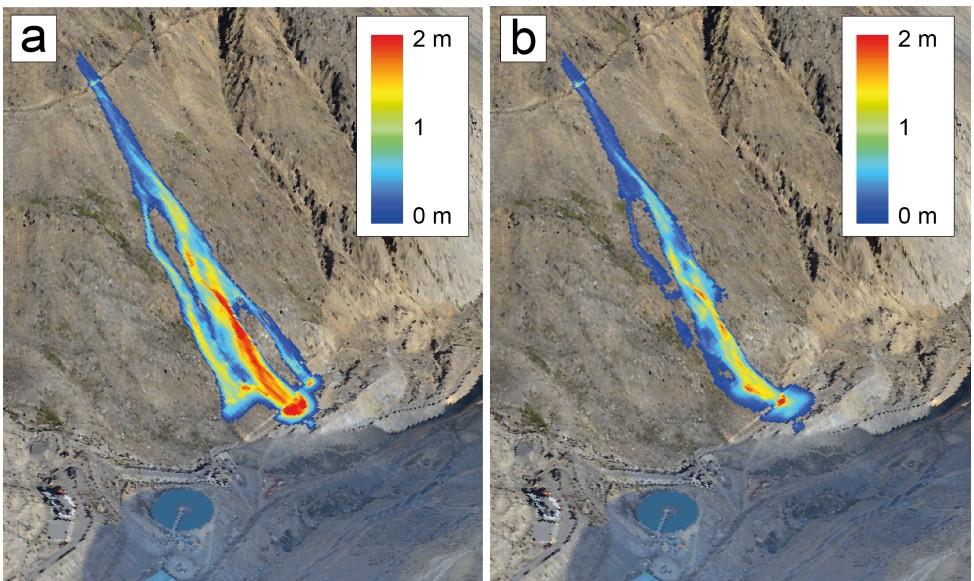

**Fig. 17.** Case study **LGW-2** simulated with (a) and without (b) lubrication effects. Without lubrication several flow arms (that were observed) are not reproduced.

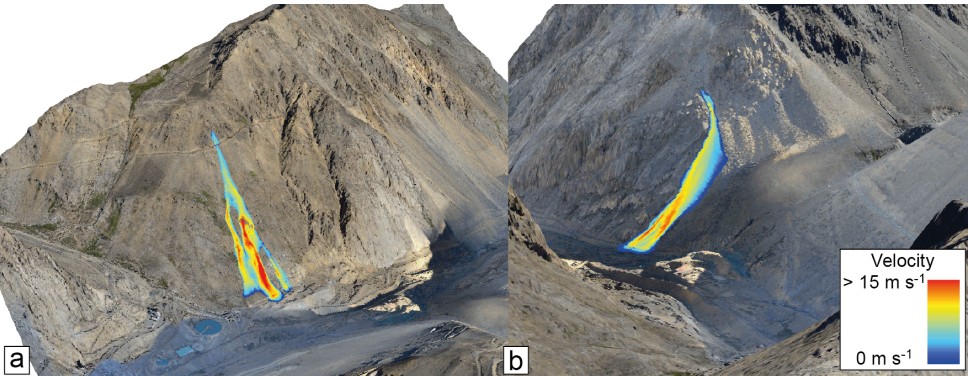

**Fig. 18.** Calculated maximum velocities of the **BN-1** avalanche (a) and of the **LGW-2** avalanche (b). Max. flow velocities reached about 15 m s$^{-1}$.