# Peer review of "Modelling wet snow avalanche runout to assess road safety at a high-altitude mine in the central Andes"

_Natural Hazards and Earth System Sciences, 2016_

## Referee Comment (RC1) · Anonymous Referee #1 · 18 Apr 2016

Wet snow avalanches are getting more important as the global warming is in progress. Based on this background, the authors have already developed the wet snow avalanche model and confirmed its validity for the several cases. Then, in this manuscript, they applied the model to the situation at the mining in central Andes. I remember the authors submitted the manuscript entitled as "Point release wet snow avalanches", that included nearly the same contents to NHESS last May. At that time I sent the comment that it was not matured yet for the publication. Comparing to the previous manuscript, this version has been improved substantially; the model is described much more in detail and the discussion part was expanded largely. As a matter of fact, almost the same explanation of the model is also found in Valero et al. (2015),

this part can be shortened if the total volume became too large. Physical processes which may happen in the wet snow avalanches are carefully taken into account for the model construction. Probably showing the way we should take. However, the main purpose of this manuscript is not the model development, but the application to the snow avalanche safety for the he specific valley. This manuscript introduces the very well documented data sets of the wet snow avalanches. Data themselves are also very valuable. So, probably it might be a good idea to split all the contents into two. First one introduces the observed wet snow avalanches and analyze the releasing conditions more in detail with obtained meteorological data and the SNOWPACK analysis. Then, the second one discusses the aspect of avalanche movement, utilizing the model output. Not only the latter approach but also the former one is essential for the comprehensive snow avalanche safety. Wet snow avalanche model shown here strongly relies on the input parameters, in particular, the snowpack properties from the releasing point to the deposition zone. In fact, i the conclusion part (line 623), authors note that the avalanche model requires the fracture depth, snow temperature, snow density and water content in the release area and along the avalanche path. Thus, at any rate, authors need to show how the SNPOWPACK worked properly enough to express the snow condition. In line 85, they say that the SNOWPACK model results were validated with field measurements (snow pits) performed by the winter operation crew. It should be shown specifically in the manuscript. When the SNOWPACK performed well, the output is applicable to determine the snow stability for the wet snow avalanche release, that is much more direct and necessary information for them. As is mentioned above, when the SNOWPACK model is utilized for this approach, the warning from this aspect will be also possible. As is also commented for the previous manuscript, I am a bit anxious whether the depth-averaged shallow water equation model is able to describe the avalanche motion precisely on the steep clip as is shown in these examples. Furthermore, since this is the continuum model, usually the flow keeps going for the long period. Is there any stopping criteria in this model and definition in the model procedures for the issues called as "dry/wet problem"? How did you determine the initial

snow depth on the avalanche track? It looks far from uniform according to the figures in the manuscript strongly depending on the topography. As you see, the re-distribution of snow by the wind will be the key issue. Needless to say, initial snow depth distribution gives the strong effect not only on the basal friction but on the erosion mass. I wonder the authors introduced ARPS as well as SNOWPACK models, and utilized to estimate the initial snow depth distribution. If it is not included in the initial condition, the following calculation sounds meaningless. As you see, snow properties, such as dry or wet, are far from satisfactory.

Following inquires need to be also addressed.

Figs. 6 to 10: SNOWPACK simulations shown here are for the releasing zone or somewhere else? Further, please describe the reason why you could conclude the SNOWPACK outcome is accurate enough for the input data of the avalanche simulation. Comparison with the pit observation should be displayed. Densities plot shown in Figs. 6 to 10 are total ones in which the water is involved in?

Fig. 10: If you could use the drone and take pictures, as you may see, the DEM for the target area and the volume of the released and deposited snow can be deduced with the software of "photoscan" . It will be very much useful when you verify the simulation output.

Table 3: This table is not necessary, because all the parameters except for ïĄąïĂăare the same for the five examples. I am wondering these nine parameters are completely enough to designate whole avalanche behavior and no other arbitrary parameters are involved in the model. No fitting parameters remained.

Line 447: " Calculated runout distances are in good agreements with the GPS measurements….." : It is probably a good idea to summarize in the table, Since the model output shown in Figs 6 to 10 are rather qualitative and detail are not clear.

Line 460: Perhaps it might be a trivial issue, but I am curious what caused such differences in the spreading? Do you have any idea how you can improve this issue?

In case of the point release avalanche, strictly speaking, it started from the specific point of the snowpack. How did you determine the initial volume? Assuming the triangle area, as is shown in the figures, maybe helpful but looks highly arbitrarily.

Fig.11: The outline of the deposits, observed by the winter operation crews, should also indicated in the figure and compared with the model output.

Fig. 12: Are there any specific reasons why the temperature shown in red becomes nearly steady sate at around -0.15 C (not 0C), even though the air temperature is substantially high and snow contains the melt water?

Fig. 13: LWC and random kinetic energy seem to correlate more or less each other. However, the relation between the friction and the LWC is not always the case. What makes such difference? Such arguments are essential in the discussion part because the models are developed in usual to recognize what is happening in the real snow avalanches; that is hard to get the information from the observations.

Fig. 17: According to my experiences, the dry snow avalanche runs longer distance with higher speed than the wet one in usual. I understand that the lubrication process plays an important role for the wet snow avalanches in particular, for the water satu­rated slush flows on the smooth rock surface. However, as is recognized on the figures the debris are not clean and apparently include the mud. That means the avalanche ran down over the ground, not the smooth snow surface. Do you have any idea what made such discrepancies with the conventional knowledge?

---

## Referee Comment (RC2) · Anonymous Referee #2 · 22 Apr 2016

The paper "Modelling wet snow avalanche runout to assess road safety at a high-altitude mine in the central Andes" is of good quality. The topic is definitely an important issue. It gives a clear overview of the actual research on wet snow avalanches explained with 5 examples. The paper consist of two parts: one part deals with wet snow modelling and the second part provides a case study from a mining road in the central Andes. In my opinion the title promises rather a case study, but in the end it is a combination of in depth explanation of the new wet avalanche modelling and the 5 examples in Chile. The material seems for me good enough for two good papers. This enables the authors to describe the new wet avalanche model precisely (with 2-3 different and simple examples for the explanation) as well as to display the applicability

of the model in the Andes in more detail in an another paper. The users would be interested to get to know how the model really works in practice (i.e. in the Andes).

I am interested especially in the interface to SNOWPACK in more detail. It seems to me that the fracture depth is insignificant as you have a growth factor (of volume) of about 20-90! What about the xi-values? You are mentioning that you rather concentrate on the current snow conditions, but what about the quite rough and rocky terrain? You are discussing avalanches of max. 10.000m$^3$, here the local terrain features (even partly without snow cover) will definitely have a strong impact on the simulations. So I would expect a few more sentences on this fact, especially as you have a 2m airborne laser scanner data available.

The purpose of this paper is the explanation of the modelling, so it would be helpful for the readers to display the simulations in a suitable scale. You have aerial photographs, GPS points and mappings, you can display the mappings together with the simulations. The congruence of the avalanche events with the modelling is good, so you can also visualize it.

Have you observed some rather random run out behaviour of the wet avalanches as turning almost in circles in the Andes as well? Have you taken into account this pattern of random turns in wet snow avalanches? What would be the consequence for the modelling? I would be very curious of a few words on this.

Line 89: An additional problem. . . -> there is no previous problem mentioned

Line 108: remove comma

Line 125 (Equ. 2): 3 times the same the equations ->the symbols "a" in the second and "w" in the third equation are missing

Line 158: . . . the notation ($^\circ$). . . -> already earlier in Equation (3) used (describe already there)

Line 165: this is an important sentence, please elaborate this more in details or give a

cite

Line 179/ 180: explain in more detail or cite

Equ. (12): Suggestion: full derivation in appendix?

Line 208: derivation -> of the thermal layer

Line 216: the symbols i, a, w are used earlier, it would be helpful already in Line 120

Line 275: hs -> see Figure 3?! (no hs in Fig. 4)

Line 297: why? Is this the result of observations? . . . more details on the observation or cite.

Line 326: when access is possible

Line 417: The avalanche -> was observed. . .

Line 447: -> then show the GPS points in the figures of the examples

Line 575: Cite this observation or give more details of the source; tan9°=0,12?? -> see Line 297/298

Line 612 to 619: -> should be in the introduction instead of the conclusion!

Figure 9: description: The model accurately . . ..?? the avalanche . . . -> missing word

---

## Author Comment (AC1) · 17 Jun 2016

**Response to Reviewer #1**                    Author: Cesar Vera Valero
nhess-2016-61        Co-authors: N. Wever, Y. Bühler, L. Stoffel, S. Margreth and P. Bartelt.
doi:10.5194/nhess-2016-61/                              Date: 17/06/2016

**Response to Reviewer # 1**

We thank both reviewers for their careful examination of the manuscript. We truly appreciate the effort and made substantial changes to the paper based on their comments.

Referee 1 raises two major points:

1. The reliability of SNOWPACK calculations

2. The influence of wind

We address point 1 (SNOWPACK) by including a supplement containing direct comparisons between snow pit measurements and SNOWPACK calculations. We additionally include one plot in the this review to demonstrate that SNOWPACK is performing with comparable skill in the Cajon del Rio Blanco valley, where the mine is located, as at the Weissfluhjoch measurement site in the Swiss Alps.

We share the same concern as the reviewer that snow transport by wind is an important factor influencing the snow distribution in avalanche slopes. However, a correct assessment would require laser-scanning, or advanced simulations using distributed simulations using for example the Alpine3D model, in combination with a meteorological model to estimate the wind fields in the mountaineous terrain. We consider this to be out of the scope of the present manuscript. We hope that future studies will continue to investigate how to drive avalanche dynamics models with simulated snow cover information, possibly including snow drift effects. We will provide a more detailed response regarding this issue below.

**Reviewer:**

Wet snow avalanches are getting more important as the global warming is in progress. Based on this background, the authors have already developed the wet snow avalanche model and confirmed its validity for the several cases. Then, in this manuscript, they applied the model to the situation at the mining in central Andes. I remember the authors submitted the manuscript entitled as Point release wet snow avalanches, that included nearly the same contents to NHESS last May. At that time I sent the comment that it was not matured yet for the publication. Comparing to the previous manuscript, this version has been improved substantially; the model is described much more in detail and the discussion part was expanded largely. As a matter of fact, almost the same explanation of the model is also found in Valero et al. (2015), this part can be shortened if the total volume became too large.

**Response:**

1. Thank you, we made a big effort to improve the paper in comparison to the original manuscript. We also believe that climate change is an important aspect of the wet snow avalanche problem. Therefore, our goal was to write a self-contained manuscript describing the avalanche model in its entirety as well as several applications. We put a special emphasis on the description of the special problems involved in modelling wet snow avalanches, including lubrication, meltwater production and moist-snow entrainment.

2. Although the introduction is long, we believe that an overview of the problems of avalanche mitigation for the mining industry deserves special attention. This was clearly no part of the Vera Valero et al. (2015) paper.

3. There are several other differences to the Vera Valero et al. (2015) paper; (1) we explicitly define the mass and volume contents of ice, air and water in the flow, (2) the friction/lubrication function has changed slightly and (3) the entrainment process is described in more detail, including the heat produced during the entrainment process (plastic collision).

**Reviewer:**

Physical processes which may happen in the wet snow avalanches are carefully taken into account for the model construction. Probably showing the way we should take

**Response:**

Yes, we agree. We consider is important to develop a physics-based model, which is not relying on non-physical calibrated parameters.

**Reviewer:**

However, the main purpose of this manuscript is not the model development, but the application to the snow avalanche safety for the specific valley. This manuscript introduces the very well documented data sets of the wet snow avalanches. Data themselves are also very valuable. So, probably it might be a good idea to split all the contents into two. First one introduces the observed wet snow avalanches and analyze the releasing conditions more in detail with obtained meteorological data and the SNOWPACK analysis. Then, the second one discusses the aspect of avalanche movement, utilizing the model output. Not only the latter approach but also the former one is essential for the comprehensive snow avalanche safety.

**Response:**

Yes, we agree: wet snow avalanche data is very valuable. Especially when it is used to validate a wet snow avalanche model. This is why we would like to keep the data and the model together. We understand the comments of the reviewer and we attempted to follow his suggestion to split the contents of the paper. However, we found it was always necessary to refer to the physics of the dynamic model. It was impossible to present features of the data without discussing some physics. This places the observations in the required framework. For us it is extremely important to first describe the physics of wet snow avalanches and then describe what we can observe in the field measurements. This approach leads to direct validation of the model.

**Reviewer:**

Wet snow avalanche model shown here strongly relies on the input parameters, in particular, the snowpack properties from the releasing point to the deposition zone. In fact, in the conclusion part (line 623), authors note that the avalanche model requires the fracture depth, snow temperature, snow density and water content in the release area and along the avalanche path. Thus, at any rate, authors need to show how the SNPOWPACK worked properly enough to express the snow condition.

**Response:** Yes. we rely on SNOWPACK to provide initial conditions and we agree that the SNOWPACK results should be verified. We address this criticism by:

- We provide the following plot 1. From the data from the automatic weather station, the snow depth measurements can be used to track whether the model simulates the snow depth, and thereby the layering resulting from subsequent snow fall events, correctly. The snow surface temperature measurements were not used to drive the model and can be used as an idependent verification of the cold content of the snowpack. It is very important for the wet snow avalanche formation process to follow the warming of the snowpack to melting point correctly, as well as the onset of meltwater percolation. The plot (Fig. 1 in this document) shows the average difference in modelled vs. measured snow height and snow surface temperature (both key values for the snowcover simulations) for the last 6 winter seasons. The root mean square error in snow height is 6.33 cm and the average

snow height difference is -1.6 cm. When we compare these values with the values obtained for the Weissflujoch measurement site where snowpack is extensively tested and calibrated (see Table 1 in Wever et al, 2015) the values are a RMSE of 4 cm for WFJ and ca. 0.9 cm for snow height difference. For the snow surface temperature, the average difference is 2.3 degC in Chile and -1.5 degC for WFJ (see Wever et al, 2015). Given the fact that the Weissfluhjoch measurement site is equipped with more and higher quality sensors and is also one of the main sites used for developing and testing the SNOWPACK model, we consider the fact that the SNOWPACK calculation errors found for the Andina mine are of similar order of magnitude as for the Weissfluhjoch site an indicator that the snowcover simulations can be used in this manuscript. The error values from the Weissflujoch can be found in N. Wever et al. 'Verification of the multi-layer SNOWPACK model with different water transport schemes', Cryosphere, 2015, 9, 2271-2293. We do not plan to put Fig. 1 in the manuscript as the manuscript is already long, but we plan to discuss the performance of SNOWPACK for the Andina mine in relation to the Weissfluhjoch measurement site in the text of the revised manuscript.

[Figure]

Figure 1: Avg. error during the whole snow season of the simulated snow height and snow surface temperature for 6 winter seasons at the used meteo station

- Additional validation was performed: we dug snowpits, made hand-profiles and entered the release zone to document the fracture heights and erosion depths (when it was possible). This provided us with actual measurements to evaluate SNOWPACK. Of course, we could only validate the model at points near the avalanche path. We recognize that SNOWPACK is not perfect, but at the same time, it provides a more accurate (and rational) description of the snowcover than a best guess by an avalanche expert located far away from the slope.

- We would also like to point out that we applied SNOWPACK in a back-calculation mode. The results of the modelling were obtained after we performed the measurements. We never optimized or re-did the SNOWPACK simulations. In a forecasting mode, the uncertainties are larger, especially with respect to the location of the starting zone.

- Finally, a recent publication (Wever, N., C. Vera Valero, and C. Fierz (2016), Assessing wet snow avalanche activity using detailed physics based snowpack simulations, Geophys. Res. Lett., 43, doi:10.1002/2016GL068428) successfully applied the SNOWPACK model for the

same meteorological station in the Andina mine, and it was found that the SNOWPACK model could be used to identify wet snow avalanche days based on the same method (i.e., analysing maximum liquid water content in one of the snowpack layers) as used in this manuscript to identify the fracture depth.

**Reviewer:**
In line 85, they say that the SNOWPACK model results were validated with field measurements (snow pits) performed by the winter operation crew. It should be shown specifically in the manuscript.

**Response:** We plan to add a supplement to the manuscript, (see an example in Fig. 2 in this document). We will provide 5 comparisons between manual snow pits in the vicinity of the meteo-station and the SNOWPACK calculations at this time in the station. Furthermore, in Table 1, the erosion depth measured at the avalanche path for the BN-1 and LGW-2 cases varies between 29 and 41 cm. This depth is what we observed as wet snow cover and it was the layer mobilized in the avalanche. We provide Figs 8 and 9 to show the results of the SNOWPACK calculations on virtual slopes. The maximum water accumulation in both cases is about 20 cm below the snow surface. This is a good agreement between measurements and SNOWPACK simulations. This simulated data was used to define the release and eroded mass. Thus, the model provides a good estimation of the location of the maximum liquid water content. For the cases CG-1, CCHN-3 and CV-1, unfortunately we were not able to access to the release area because of safety reasons.

[Figure]

Figure 2: Example of the supplement to be added to the final manuscript. We will provide the hand profiles (left) together with the SNOWPACK simulations (right) performed at the AWS, projected on the corresponding virtual slope.

**Reviewer:**

When the SNOWPACK performed well, the output is applicable to determine the snow stability for the wet snow avalanche release, that is much more direct and necessary information for them. As is mentioned above, when the SNOWPACK model is utilized for this approach, the warning from this aspect will be also possible. As is also commented for the previous manuscript

**Response:** The reviewer is right and this aspect of the work is extensively discussed in the recently published paper "Assessing wet snow avalanche activity using detailed physics based snowpack simulations" in Geophysical Research Letters. The main author is co-author in this manuscript (N. Wever). In that paper, additional data from Pyrenees, Chile and the Alps is used.

**Reviewer:**

I am a bit anxious whether the depth-averaged shallow water equation model is able to describe the avalanche motion precisely on the steep clip as is shown in these examples. Furthermore, since this is the continuum model, usually the flow keeps going for the long period. Is there any stopping criteria in this model and definition in the model procedures for the issues called as dry/wet problem?

**Response:**

- Shallow water approaches do better than one may expect based on the underlying assumptions on steep slopes. The driving forces of gravity are correctly represented; the question therefore becomes if the flow friction is valid? The primary problem is not the slope angle, but the assumption of constant flow density, which defines the flow rheology. When the avalanche flies over a steep slope this assumption is clearly in error. Here we make no such assumption and therefore the dispersion of the core, when it is flying over a steep slope is modelled. This is discussed in the results section, line. 514 to 519. Nevertheless, the RAMMS model is extensively tested, verified and used in all kind of avalanche paths with satisfactory results, showing that the shallow water approach provides useful results, for relatively low computational costs.

- Yes, we used the stopping criteria used in Christen et al. (2010). This means that when the moving mass is only 5% of the maximum moving mass, we stop the calculation. We will make sure that this is clear in the revised manuscript. This approach works well when most of the mass is stopped in flat runout zones. In any case, the Cajon del rio blanco Valley is really steep with flat valley bottom, therefore the model predicted the stopping point on all the avalanche paths correctly.

**Reviewer:**

How did you determine the initial snow depth on the avalanche track? It looks far from uniform according to the figures in the manuscript strongly depending on the topography

**Response:**

1. We use the approach presented in Wever, N., C. Vera Valero, and C. Fierz (2016), Assessing wet snow avalanche activity using detailed physics based snowpack simulations, Geophys. Res. Lett., 43, doi:10.1002/2016GL068428. The main author is co-author in this manuscript (N. Wever). We consider that the release and the erosion depths will reach till the maximum water accumulation inside the snowpack. In that paper a relation was found between the ponding depth and the avalanche size. This indicated that the estimation of fracture depth by the depth of maximum water ponding is a good estimation for release and erosion depths. On the other hand, the estimate of avalanche volume in the field are close to the volumes calculated with RAMMS. This too we consider a good

indication of the correctness of the results. We will discuss the method to determine initial snow depth in more detail in the revised manuscript.

2. We stress the fact that snow depth, temperature *and* density are necessary to define the initial and boundary conditions. In fact, all three define the total internal energy of the snowcover and therefore the energy input by entrainment. Error in density are probably the most severe. For example, an error of 1 deg temperature, results in an error of 1/273 (.4 percent) in internal energy. An error in density of 100 kg/m3 can lead to a 25 percent error in the boundary conditions.

**Reviewer:**
As you see, the re-distribution of snow by the wind will be the key issue. Needless to say, initial snow depth distribution gives the strong effect not only on the basal friction but on the erosion mass. I wonder the authors introduced ARPS as well as SNOWPACK models, and utilized to estimate the initial snow depth distribution. If it is not included in the initial condition, the following calculation sounds meaningless. As you see, snow properties, such as dry or wet, are far from satisfactory.

**Response:** Yes, we would also like to introduce the influence of wind into the model. Snow redistribution by wind can modify the fracture depth and snow conditions. However, numerical simulations of drifting snow conditions need verification with laser scanning for example. Just analyzing fracture depth at the slab boundaries will not necessarily provide a correct estimate of slab depth either. Near ridges snow can accumulate due to wind, but we assume that the snow depth is well represented by the meteorological station away from the ridges. We have several indications that this assumption is justified: the eroded snow was measured and did not reach more than 40 cm in the LGW-2 and BN-1 cases, which was predicted, see Fig 8 and 9. For the CCHN-3 and CG-1 cases we have no erosion measurements but the estimated volumes in the deposits and the calculated volumes match the measurements. This is an indication that the estimated fracture and erosion depth are well represented. See **section 3. Case studies** and figure 14. As we see the importance of this point, we will ammend the manuscript with a discussion regarding this point. We consider it one of the main future developments necessary to improve the use of snowpack models to drive avalanche dynamics models.

**Reviewer:**
Following inquires need to be also addressed. Figs. 6 to 10: SNOWPACK simulations shown here are for the releasing zone or somewhere else? Further, please describe the reason why you could conclude the SNOWPACK outcome is accurate enough for the input data of the avalanche simulation.

**Response:**
The simulations were done at the meteorological station located closest to the release areas. Using the virtual slope concept (Lehning et al, 2006) we can match the actual slope and exposition of the avalanche path. This approach has provided satisfactory forecasting results, see the earlier mentioned publication by Wever et al. (2016). As noted previously: in Table 1, the erosion depth measured at the avalanche path for the BN-1 and LGW-2 cases varies between 29 and 41 cm. This depth is what we observed as wet snow cover in the field and it was the layer mobilised in the avalanche. In Figs 8 and 9 we show the SNOWPACK calculations using the virtual slopes. The maximum water accumulation in both cases is about 20 cm below the snow surface. This is a good match to the SNOWPACK values. This simulated data was used to define the release and eroded mass. Thus, the model gave a good estimation of the location in depth of the water accumulation. As mentioned earlier, we will provide a discussion regarding this point in the revised manuscript.

**Reviewer:**

Comparison with the pit observation should be displayed.

**Response:**

We will provide a supplement to the manuscript, showing 5 snow pits performed by the winter operation team in the vicinity of the meteorological station. We have measurements in the avalanche path for the cases BN-1 and LWG-2, summarized in Table 3, which match with the SNOWPACK simulations showed figures 8 and 9.

**Reviewer:**

Densities plot shown in Figs. 6 to 10 are total ones in which the water is involved in?

**Response:**

Yes the density plotted is the total one calculated by SNOWPACK. Snowcover density profile in depth. We use as input parameter the mean slab density calculated from the point with highest water accumulation till the snow surface. We will improve the caption to make this clear.

**Reviewer:**

Fig. 10: If you could use the drone and take pictures, as you may see, the DEM for the target area and the volume of the released and deposited snow can be deduced with the software of photoscan . It will be very much useful when you verify the simulation output.

**Response:**

We used GIS techniques to deduce the entrained and deposited volumes. However, to get accurate results we need accurately georeferenced drone information to determine the deposition volumes. We tried this with the data, but we found the errors in snow covered and snow free elevation models to be too inaccurate for this kind of analysis. The best thing we could obtain from the drone was the avalanche outline and compare it with the simulation outline.

**Reviewer:**

Table 3: This table is not necessary, because all the parameters except for alpha the same for the five examples. I am wondering these nine parameters are completely enough to designate whole avalanche behavior and no other arbitrary parameters are involved in the model. No fitting parameters remained.

**Response:**

We wish to include this table because it demonstrates (1) the total number of parameters in the model (a weakness) and (2) the parameters do not change (a model strength). It will be a reference when the model is applied to other regions and therefore, for us, an important contribution to the paper. An important message to convey to the reader is that the model works without extensive parameter tuning, like the standard Voellmy model. The only parameter which we need to calibrate is $\alpha$, which depends on the roughness of the avalanche track. Once this parameter is known from previous case studies, it can be fixed for future simulations. For our case studies, however, the roughness changed between $\alpha = 0.07$ and $\alpha = 0.08$. The higher value corresponds to the steeper, twisted path and the lower value to flatter, more open slopes.

**Reviewer:**

Line 447: Calculated runout distances are in good agreements with the GPS Measurements It is probably a good idea to summarize in the table, Since the model output shown in Figs 6 to 10 are rather qualitative and detail are not clear.

**Response:**

Thank you for this suggestion, we changed the plots. Calculated run out distances with measured GPS points are now included in one plot in Figs. 6 to 10

**Reviewer:**

Line 460: Perhaps it might be a trivial issue, but I am curious what caused such differences in the spreading? Do you have any idea how you can improve this issue?

**Response:**

Spreading has do with a variety of factors. Firstly, the resolution of the terrain model. Spreading can be hindered in rough, i.e. high resolution runout zones. If the resolution is smaller than the typical size of the terrain features, the model will be able to model the lateral spreading since the steepness and aspect of your calculation cell will give you the right gravity slope parallel component and therefore after integration the avalanche propagation direction. In our case the lateral spreading calculations, errors in the avalanche deposits were less than 10% see lines 455 to 458. The resolution chosen for those calculations was 2m, Secondly, spreading is very much a function of the friction parameters, the Coulomb friction $\mu_0$ and the cohesion $N_0$. Both limit spreading.

**Reviewer:**

In case of the point release avalanche, strictly speaking, it started from the specific point of the snowpack. How did you determine the initial volume? Assuming the triangle area, as is shown in the figures, maybe helpful but looks highly arbitrarily.

**Response:**

The reviewer points out an important issue. We use this triangle as initialization for the avalanche simulation, but it is just a simple procedure to start the model. Clearly the area of this triangle has no physical relation with the real release process in the real avalanche. We use the triangle procedure to start an avalanche simulation, the depth of the triangle we get from the snowpack simulations and the area is drawn such that the upper apex corresponds to the highest release point from the real avalanche. In Fig. 14 we show that the release volume is not significant for the final avalanche volume in the simulations. Also in reality, the released mass in point-avalanches is insignificant to the total mass. The model needs a minimum initial mass to initialize the avalanche, which determines the lower extend of the triangle. In the caption of figure 14 it is now included and we will expand our explanation in the manuscript at this point.

**Reviewer:**

Fig.11: The outline of the deposits, observed by the winter operation crews, should also indicated in the figure and compared with the model output.

**Response:**

Done in figures 6 to 10.

**Reviewer:**

Fig. 12: Are there any specific reasons why the temperature shown in red becomes nearly steady sate at around -0.15 C (not 0C), even though the air temperature is substantially high and snow contains the melt water?

**Response:**

There was an error in the plotting script, to transform the scale from Kelvin to Celsius was written adding 273 and not 273,15. It is solved now. Thank you

**Reviewer:**

Fig. 13: LWC and random kinetic energy seem to correlate more or less each other. However, the relation between the friction and the LWC is not always the case. What makes such difference? Such arguments are essential in the discussion part because the models are developed in usual to recognize what is happening in the real snow avalanches; that is hard to get the information from the observations.

**Response:**

The empirical relation for $\mu$ depends on the random kinetic energy (R, fluidization) as well as the liquid water content (LWC). In Fig. 13, $\mu$ is plotted for the five case studies against changes in R and LWC. The value of $\mu$ varies with changes in R as well as changes in LWC, see eqs. (22) and (23) in the manuscript. The friction parameter correlates positively with both LWC and R. However since both act together it is difficult to see which one is dominating at each time. Fig. 13 shows clearly that at several points peaks of LWC correspond to lower

values of $\mu$. This correspondence exists for every avalanche case. It can actually happen that in steep avalanche paths high values of R dominate changes in $\mu$ in comparison to changes in LWC. In the discussion lines 572-578 we address this issue directly, writing: 'A problem with depth-averaged models is that the distribution of meltwater in the avalanche height cannot be predicted from depth-averaged calculations of avalanche flow temperature, which depends on the slope perpendicular shear profile in the avalanche core. We assume that meltwater is concentrated in a shear layer whose height is in the order of magnitude of hm'.

**Reviewer:**

Fig. 17: According to my experiences, the dry snow avalanche runs longer distance with higher speed than the wet one in usual. I understand that the lubrication process plays an important role for the wet snow avalanches in particular, for the water saturated slush flows on the smooth rock surface. However, as is recognized on the figures the debris are not clean and apparently include the mud. That means the avalanche ran down over the ground, not the smooth snow surface. Do you have any idea whatmade such discrepancies with the conventional knowledge?

**Response:**

The model calculates longer runouts in case of dry snow avalanches. Fig. 17 is plotted over a summer picture with no snow. This was not the state of the valley when the avalanches released. In all the case studies the avalanches were sliding over snow, but in the CG-1 that the last 100 elevation meters it was sliding on gravel and blocks, Fig 7a. The model calculates the $\mu$ values as a function of R and LWC, even though the dry avalanches have no LWC the R is higher. Therefore the dry avalanche can run faster and longer. In the case the avalanche is sliding on bare ground, is necessary to include a polygon with higher friction values. Otherwise the model overestimates the runout distance. In the case of the CG-1 we add a polygon in the calculation with higher friction in the area where the was not more snow.

---

## Author Comment (AC2) · 17 Jun 2016

**Response to Reviewer #2**                    Author: Cesar Vera Valero

nhess-2016-61        Co-authors: N. Wever, Y. Bühler, L. Stoffel, S. Margreth and P. Bartelt.

doi:10.5194/nhess-2016-61/                    Date: 17/06/2016

**Response to Reviewer # 2**

We thank reviewer 2 for his careful examination of the manuscript.

**Reviewer:**

The paper Modelling wet snow avalanche runout to assess road safety at a high- altitude mine in the central Andes is of good quality. The topic is definitely an impor-tant issue. It gives a clear overview of the actual research on wet snow avalanchesexplained with 5 examples. The paper consist of two parts: one part deals with wetsnow modelling and the second part provides a case study from a mining road in thecentral Andes. In my opinion the title promises rather a case study, but in the end itis a combination of in depth explanation of the new wet avalanche modelling and the5 examples in Chile. The material seems for me good enough for two good papers.This enables the authors to describe the new wet avalanche model precisely (with 2-3 different and simple examples for the explanation) as well as to display the applicability of the model in the Andes in more detail in an another paper. The users would be interested to get to know how the model really works in practice (i.e. in the Andes).I am interested especially in the interface to SNOWPACK in more detail. It seems to me that the fracture depth is insignificant as you have a growth factor (of volume) of about 20-90!

**Response:** The reviewer is right: because the initial mass is so small, the release zone height has no influences on the final simulation result. What is important is the entrainment depth. In fact, all events are more or less controlled by the snowcover properties.

**Reviewer:**

What about the xi-values? You are mentioning that you rather concentrate on the current snow conditions, but what about the quite rough and rocky terrain? You are discussing avalanches of max. 10.000m3

**Response:** In the paper *Dense avalanche friction coefficients: influence of physical properties of snow" Journal of Glaciology, 2013, 59(216), 771-782. M. Naaim found that there was a correlation between $\mu$ and the snow conditions (temperature) but he did not find a correlation between $\xi$ and snow conditions. In the model $\xi$ is function of the random kinetic energy (Bartelt, 2009) but does not depend on the liquid water content of the snow. Because the decay of random energy is higher in wet flows, the value of $\xi$ is very much different than in dry flows.

**Reviewer:**

here the local terrain features (even partly without snow cover) will definitely have a strong impact on the simulations. So I would expect a few more sentences on this fact, especially as you have a 2m airborne laser scanner data available. The purpose of this paper is the explanation of the modelling, so it would be helpful for the readers to display the simulations in a suitable scale. You have aerial photographs,GPS points and mappings, you can display the mappings together with the simulations. The congruence of the avalanche events with the modelling is good, so you can also visualize it.

**Response:**

Done it in Figs. 6 to 10. The simulations results are plotted together now with the GPS measurements.

**Reviewer:**

Have you observed some rather random run out behaviour of the wet avalanches as turning almost in circles in the Andes as well? Have you taken into account this pattern of random

turns in wet snow avalanches? What would be the consequence for the modelling? I would be very curious of a few words on this.

**Response:**

It is an interesting point. In this particular valley all the avalanche paths are steep and end at the flat valley bottom. Therefore the avalanche paths have not this smooth transition with long run outs at rather flat slope angles. The avalanche records available from the mine show the avalanche always stopping at the lowest point without doing smooth curves.

**Reviewer:**

Line 89: An additional problem – there is no previous problem mentioned

**Response:**

Changed for One problem is thank you.

**Reviewer:**

Line 108: remove comma

**Response:**

Removed. Thank you.

**Reviewer:**

Line 125 (Equ. 2): 3 times the same the equations – the symbols a in the second and w in the third equation are missing.

**Response:**

Corrected. Thank you.

**Reviewer:**

Line 158: the notation (.) – already earlier in Equation (3) used (describe already there)

**Response:**

Corrected. Thank you.

**Reviewer:**

Line 165: this is an important sentence, please elaborate this more in details or give a cite

**Response:**

Citation included [Steinkogler, W., Sovilla, B., and Lehning, M. Influence of snow cover properties on avalanche dynamics. Cold Regions Science and Technology, 97, 121-131, 2014.]

**Reviewer:**

Line 179/ 180: explain in more detail or cite

**Response:**

Citations included Buser, O., and Bartelt, P. Production and decay of random kinetic energy in granular snow avalanches. J. Glaciol., 55(189), 3-12, 2009 And P. Bartelt and O. Buser and C. Vera Valero and Y. B/"uhler. Configurational energy and the formation of mixed flowing/powder snow and ice avalanches. Annals of Glaciology, 57(71), 179-188, 2016. Added

**Reviewer:**

Equ. (12): Suggestion: full derivation in appendix?

**Response:**

As we address to the first reviewer: Thank you, we made a big effort to improve the paper in comparison to the original manuscript. We also believe that climate change is an important aspect of the wet snow avalanche problem. Therefore, our goal was to write a self-contained manuscript describing the avalanche model in its entirety as well as several applications. We placed a special emphasis on the description of the special problems involved in modelling wet snow avalanches, including lubrication, meltwater production and moist-snow entrainment.

**Reviewer:**

Line 208: derivation – of the thermal layer

**Response:**

Corrected. Thank you.

**Reviewer:**

Line 216: the symbols i, a, w are used earlier, it would be helpful already in Line 120

**Response:**

Corrected. Thank you.

**Reviewer:**

Line 275: hs – see Figure 3?! (no hs in Fig. 4)

**Response:**

in figure 3 $h_s$ denotes flow height. In figure 4 $H_w$ denotes water content within the flow measured in mm of water.

**Reviewer:**

Line 297: why? Is this the result of observations? more details on the observation or cite.

**Response:**

It is an assumption we modelled like that. We chose this expression after the observations of the references.

**Reviewer:**

Line 326: when access is possible

**Response:**

Corrected. Thank you.

**Reviewer:**

Line 417: The avalanche – was observed

**Response:**

Corrected. Thank you.

**Reviewer:**

Line 447: – then show the GPS points in the figures of the examples All the GPS measurements are now included in the Figures 6 to 10 together with the avalanche simulations.

**Reviewer:**

Line 575: Cite this observation or give more details of the source; tan9=0,12?? – see Line 297/298

**Response:**

Corrected. Thank you.

**Reviewer:**

Line 612 to 619: – should be in the introduction instead of the conclusion!

**Response:**

These lines summarize the major goals of the paper and we consider them a good introduction for the conclusions.

**Reviewer:**

Figure 9: description: The model accurately .?? the avalanche – missing word

**Response:**

Corrected. Thank you